# Carafe enables high quality in silico spectral library generation for data-independent acquisition proteomics

Bo Wen [1], Chris Hsu[1], David Shteynberg[1], Wen-Feng Zeng [2], Michael Riffle [1], Alexis Chang [1], Miranda C. Mudge [1], Brook L. Nunn [1], Brendan X. MacLean [1], Matthew D. Berg [1], Judit Villén [1], Michael J. MacCoss [1] ✉ & William S. Noble [1,3] ✉

Data-independent acquisition (DIA)-based mass spectrometry is becoming an increasingly popular mass spectrometry acquisition strategy for carrying out quantitative proteomics experiments. Most of the popular DIA search engines make use of in silico generated spectral libraries. However, the generation of high-quality spectral libraries for DIA data analysis remains a challenge, particularly because most such libraries are generated directly from data-dependent acquisition (DDA) data or are from in silico prediction using models trained on DDA data. In this study, we introduce Carafe, a tool that generates high-quality experiment-specific in silico spectral libraries by training deep learning models directly on DIA data. We demonstrate the performance of Carafe on a wide range of DIA datasets, where we observe improved fragment ion intensity prediction and peptide detection relative to existing pretrained DDA models. To make Carafe more accessible to the community, we integrate Carafe into the widely used Skyline tool.

Data-independent acquisition (DIA) has emerged as a powerful strategy for collecting liquid chromatography (LC) coupled tandem mass spectrometry (LC-MS/MS) data in bottom-up quantitative proteomics[1,2]. The growth of this acquisition strategy has been enabled by significant technical advances both in the precursor isolation specificity and sensitivity[3,4]. These advances have made DIA not only practical but preferable for the systematic and reproducible sampling of peptides for quantitative analysis. DIA produces product ion data independent of whether a precursor signal is detectable and is analogous to targeted parallel reaction monitoring (PRM). However, in contrast to PRM, DIA data is collected across a mass-to-charge (m/z) range and not on predefined target peptides. Furthermore, DIA has been enabled by a shift in computational strategy from searching MS/MS or MS2 spectra against peptide sequences to searching peptides against the LC-MS/MS data[5]. This strategy, referred to as peptide-centric searching, eliminates the assumption that each MS2 spectrum is produced by a single peptide, and most tools that implement the peptide-centric method assume that each peptide's fragment information is present in multiple continuous MS2 spectra at a retention time (RT) defined by the hydrophobicity of the sequence. Each peptide is scored against the data, often as extracted ion chromatograms (XICs), and not against specific spectra. This effectively multiplexes the MS2 spectra acquisition enabling the number of detectable peptides to exceed the number of spectra. Overall, the maturity of the hardware, combined with advances in computational methods, have facilitated systematic sampling of analytes and improved the quantitative analysis of complex peptide mixtures from protein digests.

To analyze these data, software tools benefit from the use of existing metadata in the form of "spectral libraries" for analyzing DIA data. In this setting, a "spectral library" refers to a list of peptides with

[1]Department of Genome Sciences, University of Washington, Seattle, WA, USA. [2]Department of Proteomics and Signal Transduction, Max Planck Institute of Biochemistry, Martinsried, Germany. [3]Paul G. Allen School of Computer Science and Engineering, University of Washington, Seattle, WA, USA. ✉e-mail: maccoss@uw.edu; wnoble@uw.edu

specific charge states, derived from the proteome of interest, each with estimates of the peptide's RT, fragment ion intensities, and any other information like ion mobility collisional cross-section that can be used to improve the assignment of a peptide sequence to a signal in the data. In practice, accurate estimates of these quantities are critical to achieving accurate peptide detection from DIA data[6,7]. Accordingly, significant efforts have been made to generate high quality spectral libraries, typically by making use of data generated via data-dependent acquisition (DDA)[8,9]. Best results are achieved when the spectral library is generated with the same instrument settings as those used for DIA data generation[10]. In practice, generating a comprehensive spectral library by DDA requires multiple biochemical fractions and, thus, is time-consuming and isn't particularly amendable to reuse.

One potential solution to this challenge comes from machine learning. In recent years, deep learning models have been developed that predict both peptide RT and fragment ion intensity highly accurately[11–16]. One of the major applications of these deep learning predictions is to generate in silico spectral libraries for DIA analysis[7,12,17,18]. Indeed, it has been shown that these predicted spectral libraries yield comparable or superior performance compared to empirical spectral libraries generated from DDA experiments[7,12,19], potentially obviating the need to generate a new empirical spectral library for each project.

Unfortunately, both types of spectral libraries—empirical and predicted—suffer from a mismatch between the DDA data used to generate the library and the DIA data analyzed with that library. In particular, fragment ion intensities differ systematically between these two acquisition strategies. This difference arises largely due to differences in collision energy optimization between DDA and DIA[20–22]. Specifically, in DDA, collision energy is typically optimized based on both precursor charge state and mass-to-charge ratio (m/z) for each peptide, whereas in DIA, collision energy is typically optimized based on a fixed charge state and the center m/z of an isolation window during peptide fragmentation. Consequently, the same peptide will yield different fragment ion intensities in DDA and DIA settings. In addition, the settings of LC coupled with mass spectrometry used in DDA and DIA experiments are often different when both types of data are generated on different instruments or in different laboratories, leading to differences in RTs between the two types of data that are difficult to calibrate effectively.

To address this mismatch, several methods have been developed to use DIA data in the construction of spectral libraries. For example, Searle et al. demonstrated that incorporating gas phase fractionated (GPF) DIA data information into an empirical spectral library leads to a significant boost in the number of detected peptides relative to a DDA-based library[20]. Similarly, the MSLibrarian method improves the peptide detection power of a predicted spectral library by leveraging information from DIA data[22]. In brief, MSLibrarian first applies DIA-Umpire[23] on a given DIA dataset to generate pseudo-DDA spectra and then performs a traditional database search on the pseudo-DDA spectra. Based on the identified peptides and their matched pseudo-DDA spectra, MSLibrarian then optimizes peptide fragment ion intensity prediction parameters using the Prosit machine learning model[12] and calibrates RT predictions. MSLibrarian also reduces the size of the in silico generated spectral libraries by only including proteins identified by a database search using a relaxed false discovery rate (FDR) threshold and limiting the number of fragment ions.

In the context of predicted spectral libraries, the three-stage approach adopted by MSLibrarian can be improved. First, the method used to optimize prediction parameters does not account for different peptide fragmentation strategies. Second, rather than first training a deep learning model using DDA data and then optimizing that model's prediction parameters to account for the DIA settings, it is preferable to train a fragment ion intensity prediction model using DIA data directly. However, this direct approach is complicated because DIA spectra are highly chimeric. Thus, obtaining DIA training data with accurate fragment ion intensities and without interference is difficult. In addition, most of previously published tools[7,16,22,24] for generating in silico spectral libraries generally require users to have some level of bioinformatics experience for installation and usage. In particular, deep learning-based tools depend on a complex stack of specific libraries, the installation and configuration of which can be demanding for users with limited bioinformatics expertise.

In this work, to address these challenges, we develop Carafe, a tool that generates high quality in silico spectral libraries by training directly on DIA data. Our approach uses a model architecture that is similar to the existing tool AlphaPeptDeep[16]. However, to enable training on chimeric spectra, we propose a two-pronged approach to accurately detect DIA fragment ion peaks that suffer from interference, i.e., peaks that are generated from fragmentation of two or more precursors. We then adapt our model training framework to support masking of these interfered peaks during training. We demonstrate the performance of Carafe on both global proteome and phosphoproteome datasets. To make Carafe easily accessible to the user community, we integrate Carafe into the widely used Skyline tool[25], streamlining the installation of its complex library dependencies and delivering an intuitive graphical interface that requires minimal bioinformatics expertise.

## Results
### An overview of Carafe
Carafe consists of three modules for generating an in silico spectral library tailored to a specific DIA LC-MS/MS experiment setting of interest (Fig. 1a). The first module generates training data for both fragment ion intensity and RT prediction from DIA data. The DIA data could be a single shot DIA-MS run generated from a sample from a model organism (e.g., a human cell line) under specific LC-MS/MS settings of interest. This module accepts as input peptide detection results from DIA tools in a tab-separated values (TSV) format and currently supports output produced by DIA-NN[26] and Skyline[25]. The second module is used to train the RT and fragment ion intensity prediction models using the training data generated by the first module. In practice, this step involves fine-tuning the AlphaPeptDeep pretrained models, which were previously trained using a large amount of DDA data (Methods). The third module generates the in silico spectral library tailored to the specific LC-MS/MS setting used to generate the training DIA data by using the trained models from the second module. Once the model is trained, it can be used to generate in silico libraries for any species. Currently, Carafe is capable of generating spectral libraries in Apache Parquet column-oriented data storage format, row-oriented TSV data format compatible with DIA-NN, EncyclopeDIA[20] and Spectronaut, the blib format supported by Skyline[25], and the HUPO-PSI standardized spectral library format mzSpecLib[27].

When training a fragment ion intensity prediction model using DDA spectra, the training data is typically generated by matching peptides with their DDA spectra by assuming that each DDA spectrum is generated from a single peptide. Therefore, for a given peptide spectrum match, all matched fragment ion peaks are considered to be solely contributed from the matched peptide (Fig. 1b). However, in DIA, spectra are naturally chimeric. Consequently, some of the matched fragment ion peaks may be produced by multiple peptides (Fig. 1b). Peaks with interference are often highly abundant because multiple fragment ions contribute to their intensities (Supplementary Figs. 1–4). The degree of interference in DIA spectra is often unknown and varies across experiments. Accordingly, the effect of training with interfered peaks is difficult to predict a priori. Therefore, masking out such peaks during training could help mitigate their potential adverse impact on performance. Carafe implements two distinct interference detection methods, and a peak is labeled as "shared" if either method

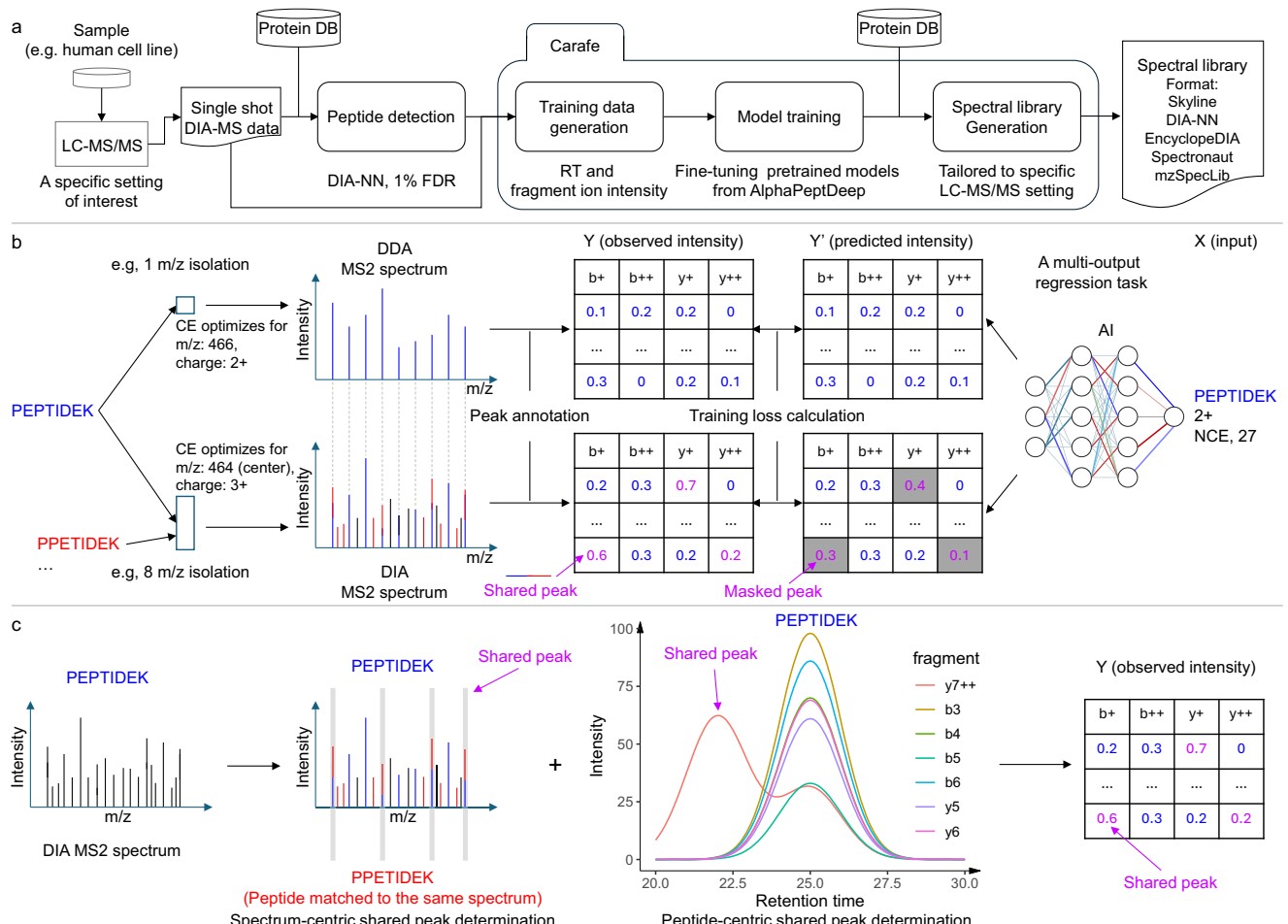

**Fig. 1 | Overview of Carafe. a** Using Carafe to generate an experiment-specific in silico spectral library for DIA data analysis. **b** The difference in fragment ion intensity model training between using DDA data (top) and using DIA data (bottom). The numbers in the tables represent illustrative (not real) observed (left) or predicted (right) fragment ion intensities, normalized to a range of 0–1. Each column represents a different type of fragment ion with a specific charge state (e.g., "b +" in the first column means a b-ion with charge 1+). Each row corresponds to the fragment ions generated by the fragmentation of a peptide at specific peptide bonds between amino acids (e.g., the first row in the first column is the b1 ion).

**c** Shared peak determination in Carafe: the spectrum-centric approach (left) and the peptide-centric approach (right). The former determines shared fragment ions by identifying fragment ions in a single MS2 spectrum that are matched to multiple detected peptides. The latter determines shared peaks through by the correlation of the extracted ion chromatograms of fragment ions from the same peptide. The two methods are complementary and are performed independently. The shared peaks determined by the two methods are combined, and that information is used for peak masking during fragment ion intensity prediction model training in Carafe. CE collision energy, NCE normalized collision energy.

detects interference (Fig. 1b–c). The first, spectrum-centric approach identifies peaks in a single MS2 spectrum that are associated with at least two different detected peptides (Supplementary Fig. 1), whereas the second, peptide-centric approach looks for peaks that show correlation with the other fragment ions for a given peptide (Fig. 1c, details in "Methods"). The second approach does not rely on detecting multiple peptides from the same MS2 spectrum to determine shared peaks (Supplementary Figs. 2–4). During model training, all shared peaks are masked out and thus do not contribute to the training loss. In principle, this peak masking training strategy could be easily adapted to work with any existing fragment ion intensity prediction framework by updating the loss function to support peak masking.

To evaluate the performance of the shared peak detection methods, we compared the Pearson correlation between observed fragment ion intensities and the fragment ion intensities predicted using the pretrained DDA model in AlphaPeptDeep for peaks detected as interfered by one method, the other, both, or neither on a TripleTOF 5600 human DIA dataset. Overall, we observed that 48.4% of detected peaks were detected as shared peaks by Carafe. Among these shared peaks, the vast majority (94.9%) were detected only by the peptide-centric method, with 2.2% detected only by the spectrum-centric

method and 2.9% detected by both methods. Scatter plots of observed versus predicted fragment ion intensities (Supplementary Fig. 5) show that the correlation is highest (0.82) for peaks detected as non-interfered by both methods, whereas the correlation is lowest (0.32) for peaks detected as interfered by both methods. This result supports the idea that peak interference can lead to reduced accuracy in fragment ion intensity prediction.

## Carafe yields accurate fragment ion intensity and RT predictions for diverse DIA datasets

To evaluate the accuracy of Carafe's fragment ion intensity predictions, we compared the performance of Carafe with the pretrained DDA model from AlphaPeptDeep, thereby testing whether the DIA fine-tuning step in Carafe is beneficial. For this analysis, we used four DIA datasets generated on three MS instrument types (TripleTOF 5600, Orbitrap Exploris 480, and Orbitrap Astral) from two vendors: two global proteome datasets, one metaproteome dataset and one phosphoproteome dataset. Each dataset contains DIA data generated from two different types of samples (a human sample and a yeast or a metaproteome sample). For each dataset, we used a single DIA run from human as training data and a single DIA run from yeast or the

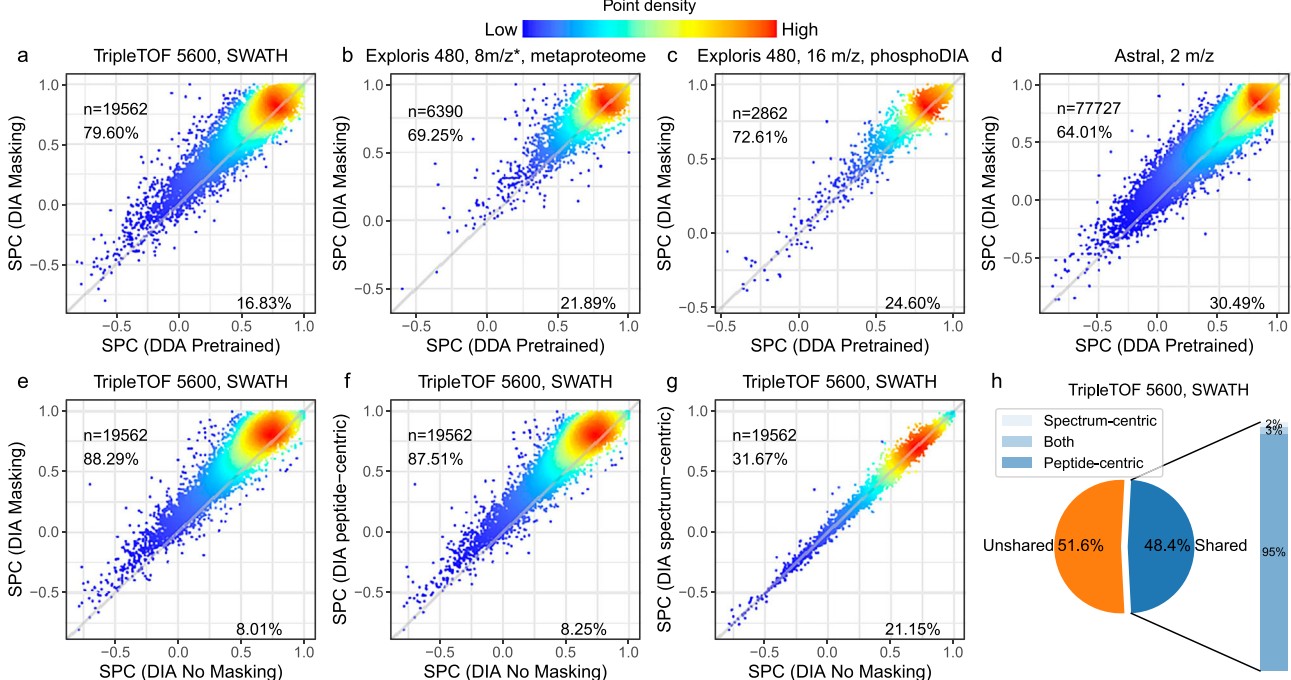

**Fig. 2 | Fine-tuning with DIA data improves fragment ion intensity prediction.** **a**–**d** Fragment ion intensity prediction performance comparison between a pre-trained DDA model (x-axis) and models trained using DIA data (y-axis) on four different datasets. SPC: Spearman correlation. **e**–**g** The performance of the shared peak detection methods in fragment ion intensity prediction. Four models were trained using the same human training data but with different peak masking strategies: no masking, spectrum-centric masking, peptide-centric masking, or both strategies. The models were evaluated on the same test TripleTOF 5600 yeast DIA dataset. The test data were generated using the same process with the training data. Each panel compares the Spearman correlation between observed and predicted fragment ion intensities for two different masking strategies. Each dot represents a peptide precursor (peptide sequence + modification + precursor charge state). In each plot, *n* is the number of precursors used in evaluation. The values are Spearman correlations between observed fragment ion intensities and predicted intensities. Only peaks determined to be non-interfered were used in the correlation calculation. The percentages on each plot indicate the proportion of spectra for which one method achieves a higher or lower correlation than the other. **h** The proportion of shared peaks detected on the TripleTOF 5600 DIA dataset. "*": staggered isolation window. Source data are provided as a Source Data file.

metaproteome as testing data. As our overall performance measure, we used the Spearman correlation between the observed and predicted fragment ion intensities, computed only with respect to unmasked peaks.

The results of this experiment show the benefits of fine-tuning with DIA data (Fig. 2a–d). For example, on the TripleTOF 5600 dataset, we observed that fine-tuning the model yielded an improved Spearman correlation for 79.60% of the peptides, with an overall median increase in correlation of 0.05. We observed that precursors with higher charge states tend to benefit more from fine-tuning (Supplementary Fig. 6). We also observed substantial rates of improvement for the other three datasets (69.25% for the metaproteome data, 72.61% for the phosphoproteomics DIA data, and 64.01% for Orbitrap Astral data). Notably, the pretrained AlphaPeptDeep DDA model only used data from Thermo Scientific and Bruker instruments but provides a good starting point when fine-tuning with data from a TripleTOF instrument, which is from a different instrument vendor SCIEX (Fig. 2a).

To further investigate the benefits provided by our peak masking methodology, we also performed an ablation experiment, in which we eliminated one, the other, or both of our two interference detection methods from Carafe. We compared the performance of the primary Carafe model with each of the three ablation models (peptide-centric only, spectrum-centric only, or no masking) by calculating the proportion of peptides with improved peak intensity predictions, as measured by Spearman correlation. As shown in Fig. 2e–g, the best performance is achieved when both methods are used. Between the two interference detection methods, the peptide-centric approach appears to be more effective than the spectrum-centric one, as evidenced by the fact that the peptide-centric method alone outperforms

the spectrum-centric method alone, and adding the spectrum-centric method to the peptide-centric method only slightly improves the performance. This observation is expected because most of the shared peaks are determined by the peptide-centric method as shown in Fig. 2h. These results suggest that the two interference detection methods are complementary.

Finally, we also confirmed the quality of our RT predictions by repeating a similar setup (train on human and test on yeast) on four DIA datasets generated using different LC settings, including three global proteome DIA datasets and one phosphoproteomics DIA dataset. As illustrated in Fig. 3, the RT models fine-tuned with DIA data showed superior performance over the pretrained DDA model, as evidenced by the improved squared correlation coefficient ($R^2$) values. The fine-tuned RT models consistently achieved $R^2$ values exceeding 0.98 across the datasets. Remarkably, the fine-tuned RT models consistently achieved much better linear correlation and tightening of RT accuracy across the datasets, while the pretrained DDA model showed non-linear correlation on some of the datasets. On two of the DIA datasets (TripleTOF 5600 and Astral), the predictions from the pretrained DDA RT model suggest that the peptides eluted very late have very different elution behavior between the data used to train the DDA RT model and the DIA data. However, after fine-tuning the pretrained DDA RT model using the human DIA data, the elution behavior was learned effectively by the model as evidenced by consistent linear correlation across the whole LC gradient.

**Carafe increases the number of peptides detected with DIA-NN**

Having established the accuracy of Carafe's predictions, we next evaluated the utility of its in silico spectral libraries on four different

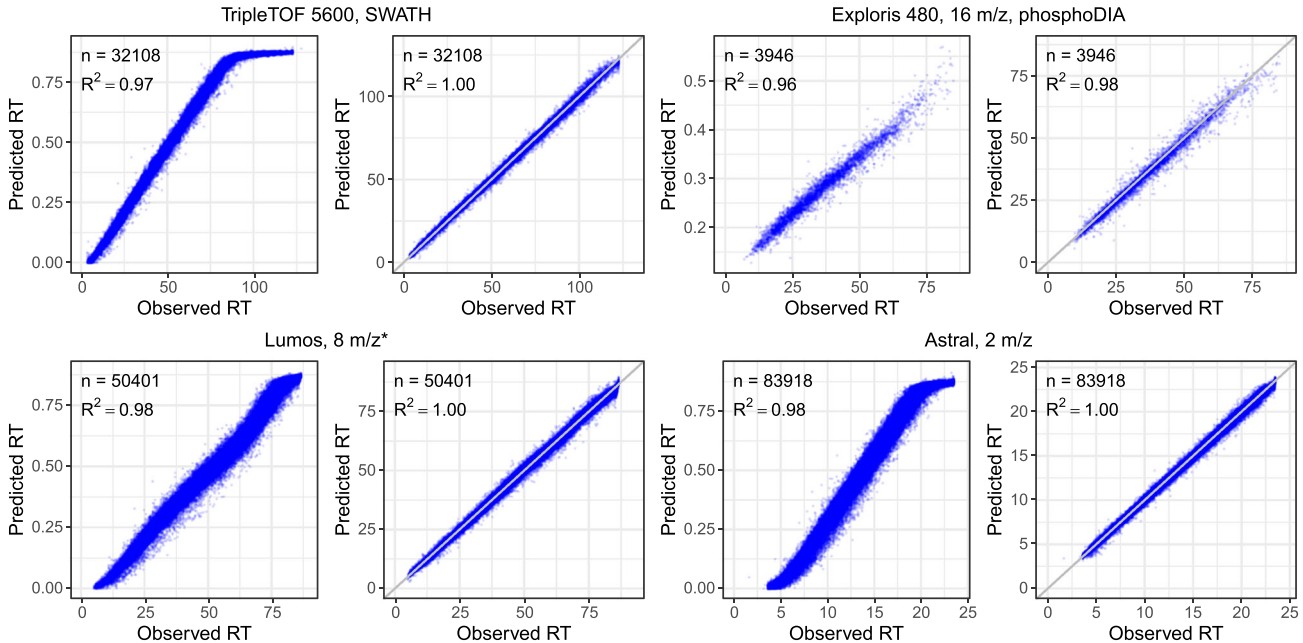

**Fig. 3 | Fine-tuning with DIA data improves retention time prediction.** Each panel plots observed RT versus RT predicted using the pretrained RT model (left panel) or fine-tuned RT model (right panel) in Carafe. On each dataset, the fine-tuned RT model was trained using the human DIA data. Both types of RT models were evaluated using the corresponding yeast DIA data. The pretrained model predicts RT on a normalized scale between 0 and 1, whereas the fine-tuned models predict RT on an absolute scale aligned with the experimental RTs from the training DIA data. "*": staggered isolation window. Source data are provided as a Source Data file.

types of DIA datasets. For each dataset, we first compared three different spectral libraries, one generated using the DDA-trained Alpha-PeptDeep model (referred to as "DDA" in Fig. 4), one generated by Carafe's fully trained model using DIA data (referred to as "MS2/RT" in Fig. 4), and one generated using DIA-NN's built-in model (referred to as "DIA-NN" in Fig. 4). The latter corresponds to using DIA-NN in its "library-free" mode. As above, we trained each model using human data and evaluated the model using the corresponding yeast or metaproteome data. In this setting, the performance measure is the total number of precursor (i.e., a combination of peptide sequence, modification and precursor charge) detected by DIA-NN at a 1% precursor level FDR threshold using the given spectral library.

The results of this experiment show that Carafe offers a consistent gain in statistical power to detect precursors relative to either of the other methods tested (Fig. 4 and Supplementary Figs. 7–10). Comparing the DDA-trained AlphaPeptDeep models to Carafe's fully DIA-trained models, we observe an increase of 5.1–38.0% in the number of detected precursors. Furthermore, when comparing with DIA-NN library free search, we identified 2.6–27.1% more precursors using Carafe fine-tuned models on the four DIA datasets. A few of the peptides gained by using Carafe's fine-tuned spectral libraries with corresponding predicted spectra and observed spectra with annotated peaks are shown in Supplementary Figs. 11–13.

Next, to better understand the source of Carafe's improved statistical power, we created two additional in silico spectral libraries: one library in which the RT prediction model is fine-tuned using DIA data and the fragment ion intensity prediction model is taken directly from AlphaPeptDeep (referred to as "RT" in Fig. 4), and a second library that is the converse (i.e., only the fragment ion prediction model is fine-tuned, referred to as "MS2" in Fig. 4). The results of this experiment (presented in Fig. 4) clearly show that both types of fine-tuning are necessary: in all four cases, the fully trained Carafe outperforms either the RT-only or MS2-only models. Furthermore, both of the semi-trained models consistently outperform the DDA-trained model. This trend is consistent across different FDR thresholds (Supplementary Fig. 14). However, the relative performance of RT-only and MS2-only differs by

dataset, suggesting that both types of predictions are necessary and neither is consistently more important than the other. To investigate the utility of fine-tuning of fragment ion intensity predictions, we compared the precursors detected using the Carafe fine-tuned library (referred as "MS2") and the precursors detected using the spectral library generated without fine-tuning (referred as "DDA") on the TripleTOF 5600 dataset. As shown in Supplementary Fig. 15, we found that the precursors gained by fine-tuning were those with low Spearman correlation in the pre-trained model, and that these precursors showed a larger increase in Spearman correlation after fine-tuning than the shared or lost precursors. The median Spearman correlation improvement (0.05) for the shared precursors was similar to that on the test dataset we showed previously. To further investigate the utility of fine-tuning of fragment ion intensity predictions, we generated a spectral library using Carafe without peak masking on the TripleTOF 5600 dataset. A total of 35,134 precursors were detected using DIA-NN with this library, which is 2.0% lower than the number of precursors (35,837) detected with the library generated using Carafe with peak masking. We repeated this analysis on the three other datasets used in Fig. 4 and found that the number of precursors detected using DIA-NN with the library generated without peak masking was 0.6% lower on the phosphoDIA dataset, 2.4% lower on the reCID dataset, and 7.5% lower on the metaproteome dataset. These results suggest that peak masking provides a modest but consistent improvement in peptide detection. We also compared the performance of fine-tuning of fragment ion intensity prediction with normalized collision energy (NCE) calibration (see "Methods"). In brief, following the NCE calibration procedure in a recent study[28], for a given dataset, a spectral library was generated using the pretrained DDA model from AlphaPeptDeep with the optimal NCE determined using the corresponding training data. As shown in Supplementary Fig. 16, 3.7–9.1% more precursors were detected using fine-tuning than NCE calibration on three DIA datasets. These results suggest that fine-tuning of fragment ion intensity predictions is more effective than NCE calibration.

Notably, to demonstrate the generalizability of Carafe, we generated a DIA dataset using resonance-type collision-induced

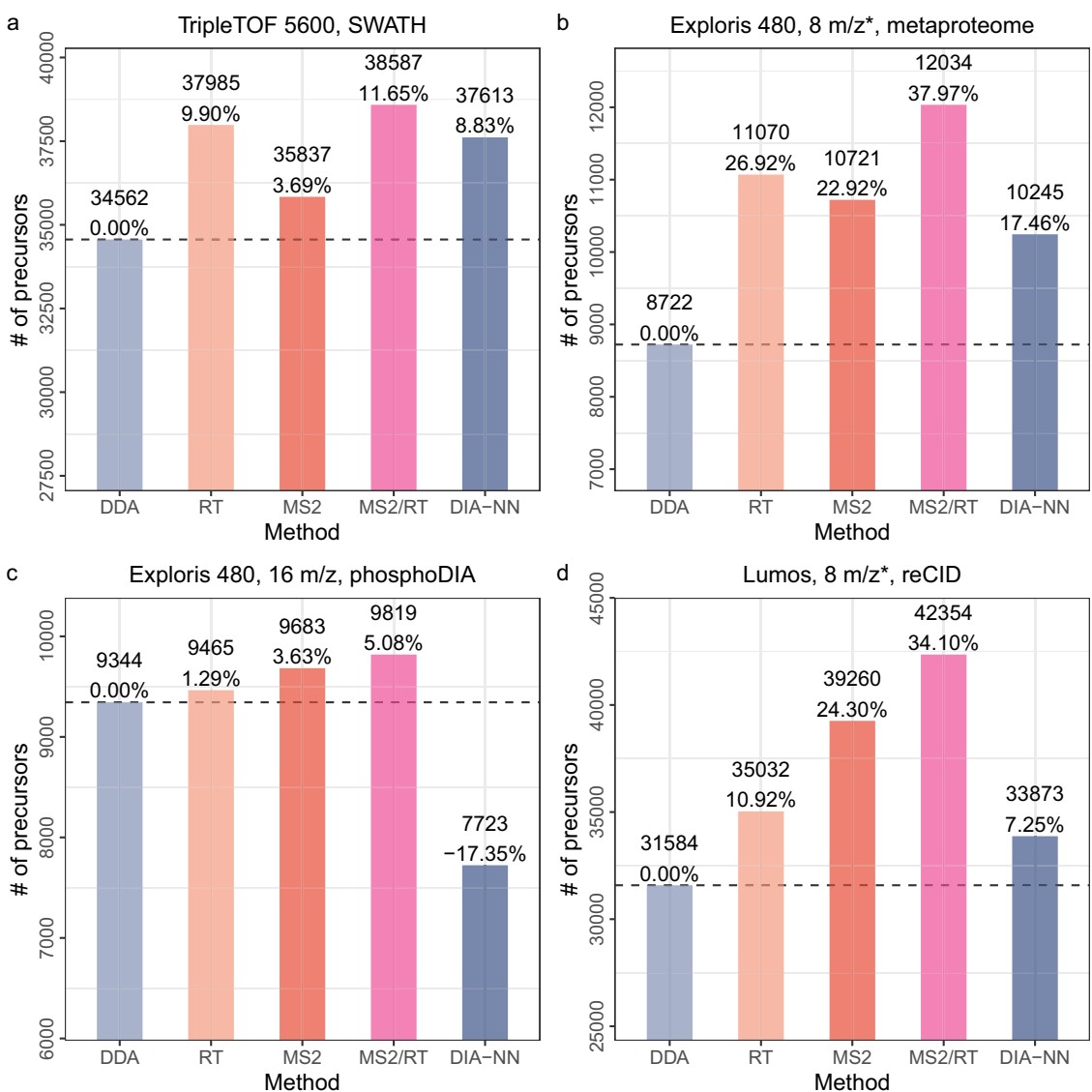

**Fig. 4 | Fine-tuning fragment ion intensity and RT prediction models improves peptide detection on DIA datasets.** Comparing in silico spectral libraries generated by different methods on four different DIA datasets. **a** A yeast dataset generated on a TripleTOF 5600 instrument. **b** A metaproteomics dataset generating on an Exploris 480 instrument. **c** A yeast phosphoproteome DIA dataset generated on an Exploris 480 instrument. **d** A yeast dataset generated using resonance-type collision-induced dissociation (reCID) fragmentation method on a Lumos instrument. Each dataset includes human DIA data as training data which was generated using the same LC and MS settings as the test yeast or metaproteome data. For the bar plots, for each dataset, the number of precursors accepted at the 1% precursor-level FDR are shown. The number on each bar is the percent improvement when comparing to the library generated using the pretrained DDA models. "*": staggered isolation window. For x-axis, DDA: spectral library generated using the pretrained DDA model from AlphaPeptDeep; MS2: spectral library generated using a fine-tuned fragment ion intensity prediction model but the pretrained RT model from AlphaPeptDeep; RT: spectral library generated using a fine-tuned RT prediction model but the pretrained fragment ion intensity model from AlphaPeptDeep; MS2/RT: spectral library generated using fine-tuned fragment ion intensity and RT prediction models; DIA-NN: spectral library generated using DIA-NN.

dissociation (reCID) fragmentation method on a Lumos instrument for both a human cell line sample and a yeast sample. The reCID method is expected to generate spectra with very different fragment ion intensity patterns compared with spectra generated using the widely used beam-type CID method (HCD)[29]. In this analysis, all other datasets were generated using HCD. As shown in Fig. 4d, Carafe significantly outperforms both the DDA-trained model and the DIA-NN built-in model. Fine-tuning the fragment ion intensity prediction model alone on the human DIA data improves the number of detected precursors on the yeast DIA data by 24.3% compared with the pretrained DDA model.

To further investigate the quality of the spectral libraries generated by Carafe, we evaluated the FDR control at the precursor level using DIA-NN with different spectral libraries on the TripleTOF 5600

human DIA dataset (Fig. 5a). We did this employing an entrapment strategy, using two different techniques that yield an upper bound and a lower bound, as previously described[30]. We observed that the FDR control is similar between the DDA-trained models (Fig. 5b–c) and Carafe fine-tuned models (Fig. 5d–f). In particular, when we analyzed the same human DIA data and compared the estimated false discovery proportion (FDP) using the upper bound method for DIA-NN with three different spectral libraries at 1% FDR threshold reported by DIA-NN, we obtained estimates of 1.07% using the spectral library generated by the pretrained DDA models (Fig. 5b), 1.00% using the spectral library generated by DIA-NN (Fig. 5c), and 1.09% using a Carafe fine-tuned spectral library trained on a yeast DIA run from the same experiment (Fig. 5d). Furthermore, the estimated FDP was 1.07% using

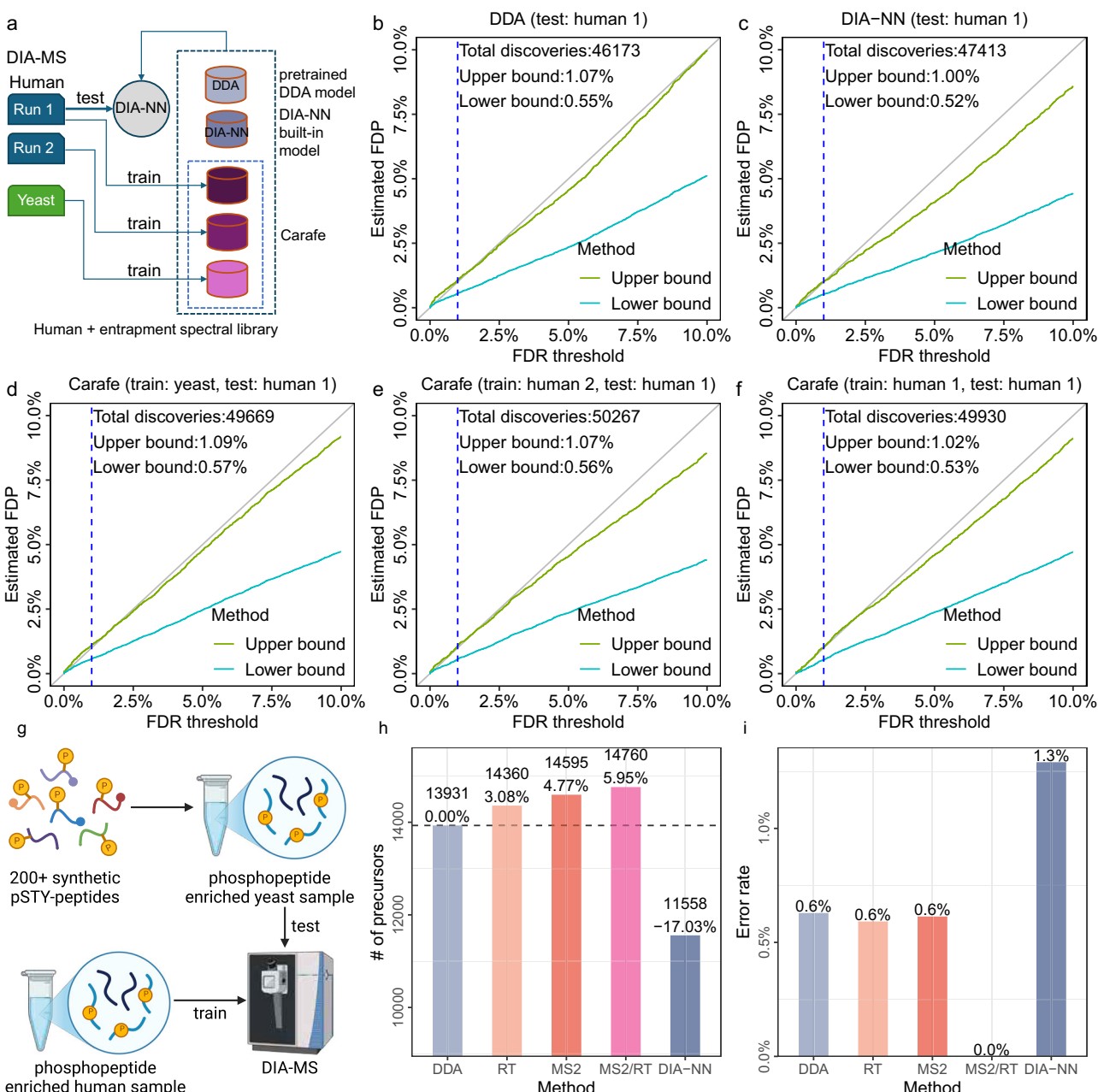

**Fig. 5 | FDR control and false localization rate evaluation.** FDR control evaluation was performed using an entrapment strategy on the TripleTOF 5600 human DIA dataset using five different in silico spectral libraries (**a**): **b** a spectral library generated using the pretrained DDA models from AlphaPeptDeep; **c** a spectral library generated using DIA-NN's built-in model; **d** a spectral library generated using Carafe with models fine-tuned on a yeast DIA run from the same experiment; **e** a spectral library generated using Carafe with models fine-tuned on a replicate human DIA run from the same experiment, and **f** a spectral library generated using Carafe with models fine-tuned on the same human DIA run. The x-axis shows different FDR thresholds reported by DIA-NN, while the y-axis shows the estimated FDPs using the upper bound and lower bound methods. The dashed vertical line is at the 1% FDR threshold, as are the numbers reported in the text in the figure. False localization rate evaluation was performed using a yeast phosphoproteome DIA dataset generated with a set of spiked-in, synthetic phosphopeptides (**g**). A human phosphoproteomics DIA MS run generated in the same study was used as training data to generate fine-tuned spectral libraries by Carafe. The data was analyzed using DIA-NN with different spectral libraries: **h** phosphopeptide precursors detected using different spectral libraries; **i** false localization rate evaluation using synthetic phosphopeptides detected with different spectral libraries. **g** was created in BioRender. Wen, B. (2025) https://BioRender.com/ewebbkb. Source data are provided as a Source Data file.

a Carafe fine-tuned spectral library trained on a replicate DIA run (Fig. 5e), while the estimated FDP was 1.02% using a Carafe fine-tuned spectral library trained on the same DIA run (Fig. 5f). These results suggest that neither data leakage nor overfitting is an issue in Carafe library generation with fine-tuned models.

To further validate the quality of the spectral libraries generated by Carafe for phosphoproteome DIA data analysis, we estimated the false localization rates when analyzing a yeast phosphoproteome DIA dataset generated with a set of spiked in, synthetic phosphopeptides using DIA-NN with different spectral libraries (Fig. 5g, "Methods"). As shown in Fig. 5h, using stringent cutoffs (PTM.Q.Value ≤ 0.01 and PTM.Site.Confidence ≥ 0.75), we identified 27.7% more phosphopeptides with Carafe's fully fine-tuned spectral library than with the library generated by DIA-NN. Consistent with our previous analysis, the fully

fine-tuned Carafe library outperformed the libraries generated using the pretrained DDA model and the partially fine-tuned models. To evaluate the false localization rate, we calculated the proportion of detected synthetic phosphopeptide precursors with incorrect site localization. As shown in Fig. 5i, the false localization rate was 0.0% using the fully fine-tuned Carafe library, whereas the false localization rate was 1.3% using the library generated by DIA-NN. Similar results were observed using a different replicate from the same experiment (Supplementary Fig. 17). These results provide evidence that Carafe is capable of generating high quality spectral libraries for phosphoproteome DIA data analysis.

### Carafe produces improved fragment ion intensities and RTs relative to libraries derived from GPF DIA data

To improve the quality of spectral libraries for DIA data analysis, the search tool EncyclopeDIA[20] implements an alternative approach to incorporating DIA data into a spectral library. In brief, a set of narrow window GPF DIA runs are first searched against an in silico spectral library created by using a tool such as Prosit[12] to generate a chromatogram library that is tailored to a specific LC-MS/MS setup. In addition to the RTs and fragment ion intensities stored in a typical spectral library, this chromatogram library also includes information about peak shapes and known interferences. Notably, this chromatogram library is substantially smaller than a standard proteome-wide spectral library, because the chromatogram library only contains entries for peptides that can be confidently detected from the GPF DIA data. The chromatogram library is then used to analyze wide window DIA data for peptide quantification. The wide window DIA data is typically generated using a wide isolation window scheme with the same LC settings and the same MS instrument as the GPF DIA data. Empirical evidence suggests that using GPF DIA data in this way yields improved correlation to spectra in DIA data, relative to correlating DDA data to spectra in DIA data[20].

We aimed to compare Carafe's spectral libraries, which are produced by models fine-tuned using DIA, against the EncyclopeDIA's DIA-based chromatogram libraries. Accordingly, we generated four different types of spectral libraries for a single DIA dataset generated on a Q Exactive HF-X instrument. The first two libraries are in silico spectral libraries produced using Prosit and Carafe models, respectively. The next two libraries are DIA-based chromatogram libraries created by searching a set of GPF DIA runs using EncyclopeDIA against the Prosit-predicted in silico library ("Prosit + GPF") and against the Carafe-predicted in silico spectral library ("Carafe + GPF"). We then searched a set of wide window DIA runs from this dataset against each of the four spectral libraries using EncyclopeDIA.

Our results suggest that the Carafe library improves upon the Prosit library. We observed that, subject to a 1% peptide-level FDR threshold, searching using the Carafe library yielded 22.6% more peptides than searching using Prosit (Fig. 6a). Similarly, between the two GPF run-derived libraries, Carafe + GPF yielded 3.2% more peptides than Prosit + GPF.

On the other hand, we also observed that Prosit+GPF outperformed Carafe alone slightly, yielding a 1.6% increase in the number of detected peptides. In practice, Prosit + GPF differs from the Carafe library in two ways. First, Prosit + GPF contains only those peptides that can be confidently detected in the GPF data (44,903 precursors), whereas the Carafe library contains all peptides in the reference proteome (661,012 precursors). Second, Prosit + GPF contains two types of information—peak shape and known interferences—that are absent from the Carafe library. To isolate the effects of each of these differences, we created two additional libraries: "Carafe-reduced" is a reduced version of the Carafe library that only contains the peptides that also appear in the Prosit + GPF library and "Prosit + GPF-reduced" is the same as Prosit + GPF, except that the peak shape and known interference information is eliminated from the library.

We hypothesized that the difference in library sizes largely accounts for the observed difference in statistical power between Prosit + GPF and Carafe. Indeed, comparing Prosit + GPF and Carafe-reduced, we find that 4.2% more peptides were identified using the Carafe-reduced library. Furthermore, as shown in Fig. 6a, removing peak shape and known interference information from the Prosit + GPF library only slightly reduced the number of identified peptides. These results indicate that the quality of predicted RTs and fragment ion intensities in the Carafe library is higher than the quality of RTs and fragment ion intensities derived from GPF DIA data.

We repeated this analysis on three other DIA datasets generated using three different instruments from two different mass spectrometer vendors. As shown in Fig. 6b–d, comparing Prosit+GPF and Carafe-reduced libraries, we find that 18.5%, 6.6% and 0.7% more peptides were identified using the Carafe-reduced library on the TripleTOF 5600, Orbitrap Astral and Orbitrap Fusion Lumos datasets, respectively. Remarkably, the Carafe library outperforms the Prosit +GPF library on both the TripleTOF 5600 and the Astral datasets.

Taken together, these results demonstrate the superior quality of Carafe libraries fine-tuned with DIA data in comparison to DIA-based chromatogram libraries, while also suggesting that using GPF data to limit the number of peptides in the library is helpful in some cases.

### Quantification evaluation

To evaluate the quantification precision and accuracy of using Carafe fine-tuned spectral libraries, we analyzed the LFQbench dataset[31] using Carafe's fine-tuned library and four other spectral libraries. As shown in Fig. 7a, using the Carafe fine-tuned library we identified 21.45% more precursors compared with the library generated using the pretrained DDA models and 10.02% more precursors compared with the library generated using DIA-NN's built-in model. These results are consistent with our previous analysis, suggesting that both the fragment ion intensity and RT fine-tuning are useful for improving the number of precursors detected.

Next, we assessed quantification precision by comparing the Carafe fine-tuned library with two other libraries: one generated using the AlphaPeptDeep pretrained DDA model (hereafter "DDA") and the other generated using the DIA-NN built-in model (hereafter "DIA-NN"). We calculated the coefficient of variation (CV) for precursor intensities across three replicates for each sample (samples A and B). As illustrated in Fig. 7b, the Carafe fine-tuned library exhibited slightly lower median CVs, indicating improved precision at the precursor level relative to the DDA and DIA-NN libraries.

We next compared the quantification accuracy using Carafe's fine-tuned library versus the other two spectral libraries. We compared the estimated precursor ratios between sample A and sample B to their expected ratios (1:1 for human, 10:1 for yeast and 1:10 for E.coli). As shown in Fig. 7c–e, we observed that the quantification accuracy using the Carafe fine-tuned spectral library is comparable to both the DDA and the DIA-NN libraries.

To further assess the quantification performance of the Carafe fine-tuned library, we quantitatively compared it with the DIA-NN built-in library using an EGF-stimulated HeLa phosphoproteome DIA dataset from a previous study[32]. We first compared the phosphopeptide precursors detected using different libraries. The fully trained Carafe library outperforms all other libraries (Fig. 7f), which is consistent with our previous analysis in Figs. 4c and 5h. We then focused on the differentially expressed phosphopeptides identified using the Carafe library and the DIA-NN library. As shown in Fig. 7g, the Carafe fine-tuned library identified 20% more differentially expressed phosphopeptides than the DIA-NN library. Notably, as shown in Fig. 7h, the Carafe library detected 14 EGFR phosphosites, compared with only 8 identified by the DIA-NN library. Among the 14 EGFR phosphosites detected with Carafe, 7 were differentially expressed between the EGF-stimulated and control samples, whereas only 4 of the 8 EGFR

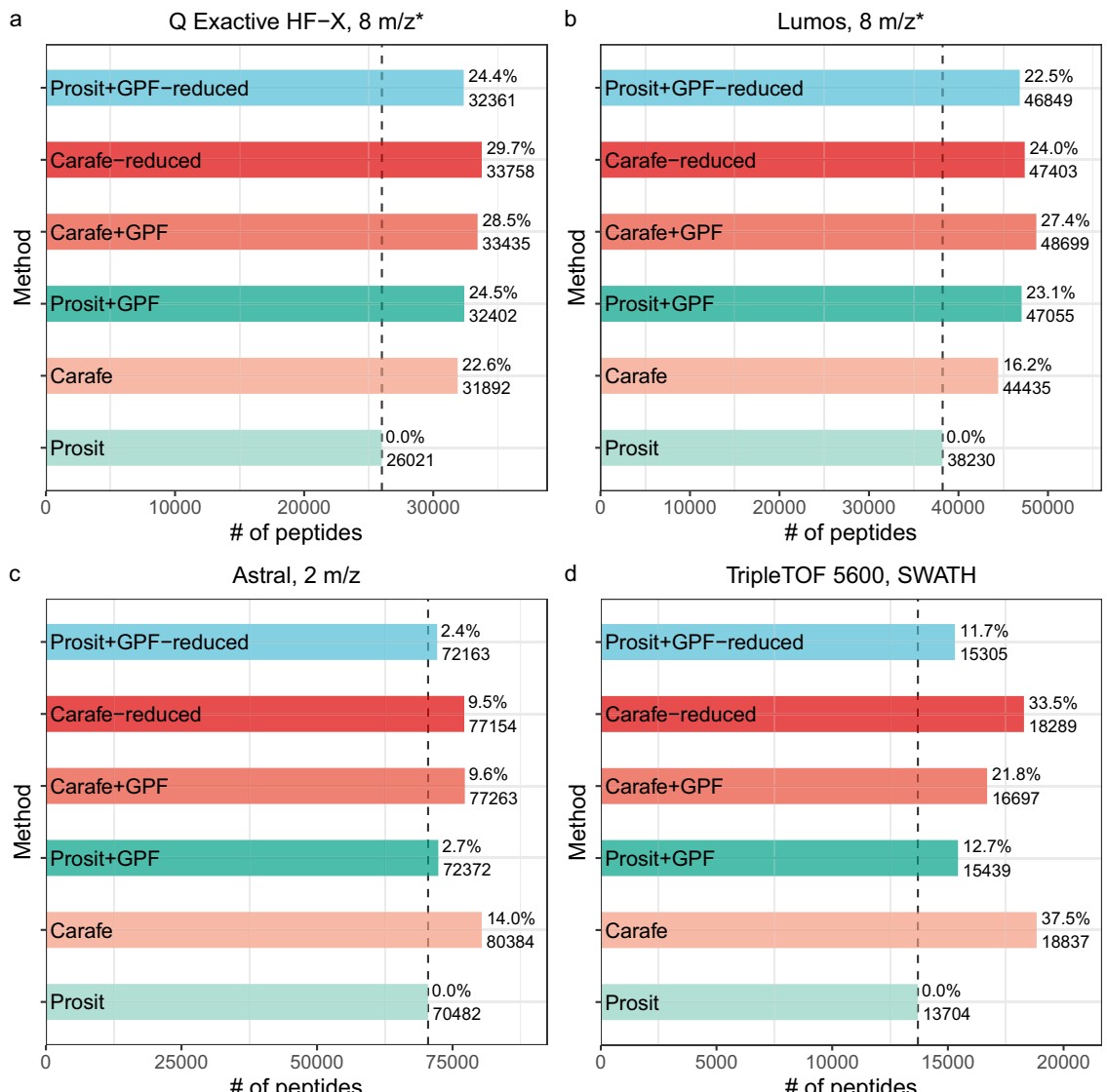

**Fig. 6 | Comparing Carafe generated libraries with DIA-based chromatogram libraries.** Peptides identified using different spectral libraries on **a** a dataset generated using a Q Exactive HF-X instrument, **b** a dataset generated using an Orbitrap Fusion Lumos instrument, **c** a dataset generated using an Orbitrap Astral instrument and **d** a dataset generated using a TripleTOF 5600 instrument. For each dataset, the number of peptides accepted at the 1% peptide-level FDR is shown. The ratio on each bar is the percent improvement when compared to the library generated using the Prosit model. "*": staggered isolation window.

phosphosites identified using the DIA-NN library were differentially expressed. Importantly, all three of the uniquely detected differentially expressed phosphosites were also reported in the original study[32]. Given that EGF-stimulated cells are known to exhibit a marked increase in EGFR phosphorylation, these results demonstrate that the quantitative measurements of the newly identified EGFR phosphopeptides using the Carafe library are in agreement with biological expectation.

## Discussion

We have demonstrated that Carafe enables high-quality in silico spectral library generation tailored to specific LC-MS/MS setups by training deep learning models directly on DIA data. Carafe provides an improvement in peptide detection on a wide range of DIA datasets generated from different mass spectrometer vendors. Compared with spectral libraries generated using DDA data trained models, Carafe's libraries improve the number of detected peptides by up to 38.0% using DIA-NN and by up to 37.5% using EncyclopeDIA. Furthermore, our results suggest that the quality of Carafe's predictions is higher

than the quality of RTs and fragment ion intensities derived from GPF DIA data. Fine-tuning on DIA data for in silico library generation raises the issue of managing shared fragment ions when constructing training data for fragment ion intensity prediction. To address this, we apply an interference peak detection method used for training data generation and a peak masking strategy used in model training to handle naturally chimeric spectra in DIA data. Although the benefit of masking may be modest under conditions where the true fraction of shared peaks tends to be low and the complexity of the MS data under analysis is not high, the actual degree of interference in DIA spectra is typically unknown and can vary substantially from one experiment to another. Consequently, training on unmasked, interfered peaks carries unpredictable effects on model performance, making it advisable to include explicit interference detection and peak masking during model training to ensure robust predictions across diverse DIA LC-MS/MS setups. In principle, Carafe's peak masking training strategy could be easily adapted to work with any existing fragment ion intensity prediction framework by updating the loss function to support peak

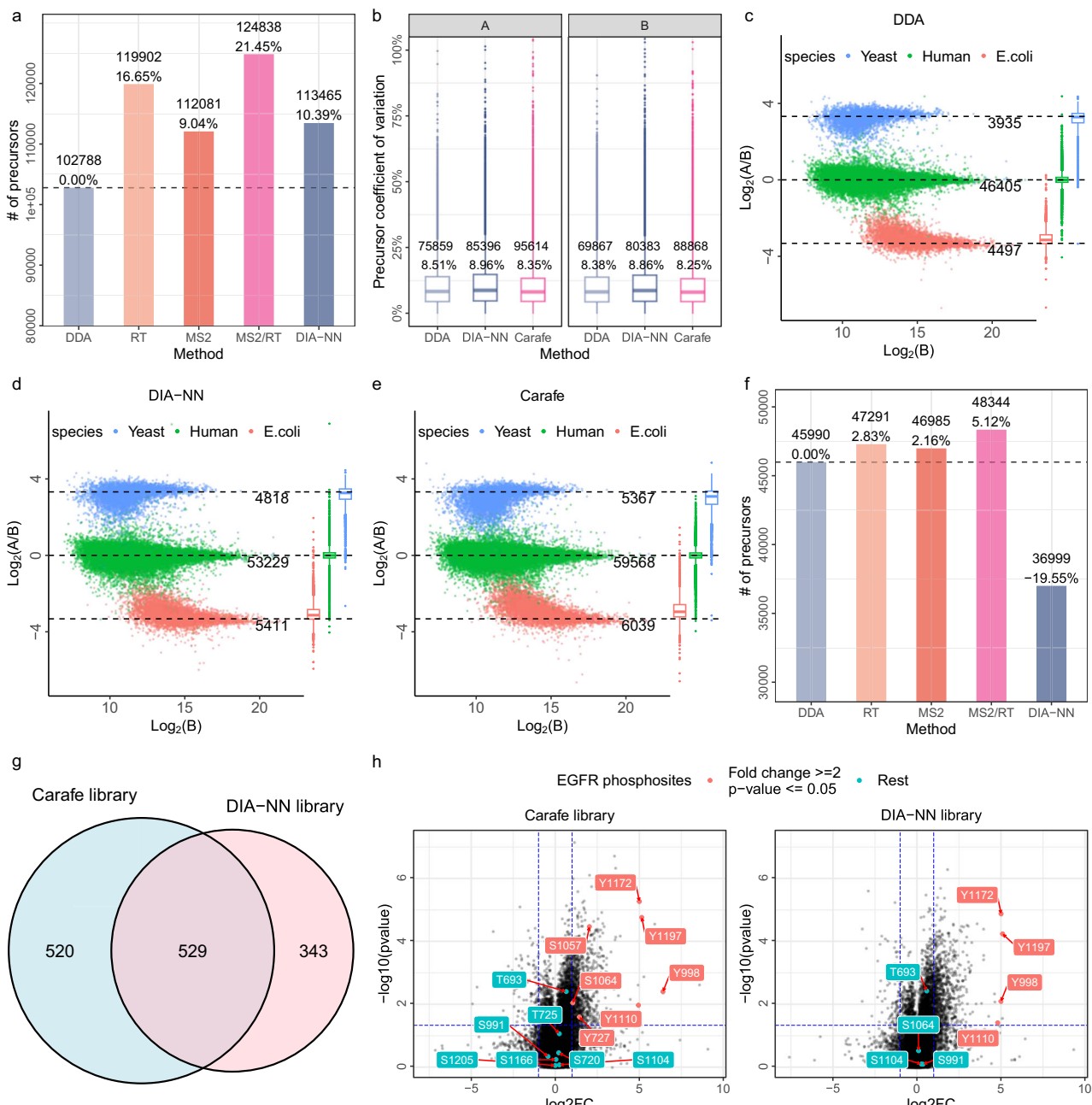

**Fig. 7 | Quantification evaluation of Carafe with DIA-NN.** Quantitative evaluation using the LFQBench DIA dataset (two samples: A and B, three technical replicates each): (**a–e**). **a** Comparing precursors detected using five different in silico spectral libraries on the LFQbench dataset. **b** The distribution of CVs of precursor intensity on the LFQbench dataset. Numbers in the first row of the boxplots are the numbers of precursors quantified. Numbers in the second row are the median CVs. **c–e** Log-transformed ratios (log2(A/B)) of precursors over log-transformed intensities of sample B for the DDA, the DIA-NN and the Carafe fine-tuned (MS2/RT) spectral libraries on the LFQbench dataset. Dashed black lines indicate the expected log2(A/ B) ratios: 1:1 for human, 10:1 for yeast and 1:10 for *E.coli*. The number under each line is the number of precursors quantified from a species used in the plot. The boxplot on the right side of each plot shows the distribution of the log2(A/B) ratios.

Quantitative evaluation using an EGF-stimulated HeLa phosphoproteome DIA dataset: (**f–h**). **f** Comparing phosphopeptide precursors detected using five different in silico spectral libraries. **g** Comparing differentially expressed phosphopeptide precursors detected using Carafe fully fined-tuned and DIA-NN libraries. **h** Volcano plots of phosphosites between EGF stimulation and control. EGFR phosphosites were labeled. The differentially expressed EGFR phosphosites were highlighted in red. The $p$ values were calculated using a two-tailed $t$-test on the log2 transformed quantification values without multiple testing correction. For boxplots, centerline indicates the median, the lower and upper hinges correspond to the first and third quartiles, whiskers indicate the 1.5 interquartile range and points indicate outliers. Source data are provided as a Source Data file.

masking. Thus, our method is general and can be implemented into DIA search engines to improve peptide detection by generating experiment-specific, high-quality in silico spectral libraries.

With fine-tuning (often referred to as transfer learning in proteomics), we show that peptides detected on a single human cell line MS run are sufficient to train high quality models for experiment-

specific spectral library generation for samples from different organisms and that this strategy generalizes well across different instruments and settings. This approach will remove the burden of generating a new empirical spectral library for a new instrument or an instrument with different MS settings (such as different fragmentation methods). The proposed method is also fast to run on a CPU since the

training data from a single MS run is small. We also show that fine-tuning of fragment ion intensity predictions is more effective than a less resource-intensive NCE calibration across different datasets. Notably, we do not observe substantial differences in spectral entropy across different NCE values. This trend likely reflects the fact that while the pretrained model was trained on data from multiple instruments, only one type of instrument (Lumos) contributed data with varying NCEs during training. However, the datasets used in our NCE analysis were acquired on different instrument types that were not represented in the NCE-diverse training data. Consequently, the model may be less sensitive to NCE variations for these specific instrument configurations, potentially limiting the representativeness of our NCE calibration results. To generate an experiment-specific in silico spectral library using Carafe, we recommend generating a single MS DIA run with the same LC and MS instrument settings as the DIA runs to be analyzed using a complex cell lysate from a model organism, such as a yeast or a human cell line sample. Thereafter, the user can fine-tune Carafe models using peptides detected from the single MS run. We also explored the possibility of using one of the DIA MS runs under study, which were all generated from the same species, to fine-tune the model in Carafe for library generation, and later using the fine-tuned library to analyze the DIA MS data from the same study. The potential for data leakage when generating the fine-tuned library directly using one of the DIA runs from the same study might introduce a bias, allowing the peptide detection tool under using to distinguish targets from decoys and thus compromise FDR control. However, we did not observe appreciable differences in FDR control when using Carafe's fine-tuned libraries generated in this way versus libraries generated using pretrained DDA models.

Because spectral libraries are essential for DIA data analysis, we expect that Carafe will be a valuable tool for the proteomics community. To make Carafe easily accessible to the broader community, we have integrated Carafe into the widely used Skyline tool, which automatically resolves and installs all required dependencies, obviating the need for manual environment setup, and presenting all functionality through an intuitive graphical interface. In the future, we plan to further improve the speed of Carafe when running on a CPU platform. We also plan to support more instruments such as the DIA data generated from Stellar and timsTOF instruments.

## Methods

### Deep learning models
For fragment ion intensity prediction we used the transformer framework implemented in AlphaPeptDeep[16], with modifications to support peak masking. The model consists of an embedding layer, a positional encoder layer, and four transformer layers followed by two fully-connected layers. The inputs to the embedding layers include amino acid sequences, modifications, and metadata including peptide charge state, NCE, and instrument type. These inputs are encoded separately and are concatenated for the following layers. The output of the last layer for each peptide is a two-dimensional matrix in which each row represents a backbone position along the peptide and each column represents a fragment ion type with a specific charge state (e.g., b1+). To accommodate shared fragment ions, we modified the AlphaPeptDeep loss function to mask selected peaks. With peak masking enabled, the masked peaks do not contribute to the overall model loss, i.e., the losses calculated from shared peaks are ignored. We started with a pretrained AlphaPeptDeep model, which had been trained using large-scale DDA data using model weights downloaded from https://github.com/MannLabs/alphapeptdeep. We then fine-tuned this model using a training set derived from DIA spectra. The training parameters were set as follows: epoch, 20; warmup epoch, 10; learning rate, 0.0001; batch size, 512. L1 loss was used for training. Both MS instrument type and NCE were extracted from the corresponding training DIA data and used as metadata inputs to the model. For the

DIA data generated using a TripleTOF 5600 instrument, the instrument type metadata was set as "Lumos" and the NCE was set to 27 during model training and prediction. For the DIA data generated using an Orbitrap Astral instrument, the instrument type metadata was set as "Lumos".

For peptide RT prediction, the hybrid convolutional neural network and bidirectional long short-term memory (BiLSTM) network model from AlphaPeptDeep was used. We fine-tuned AlphaPeptDeep's pretrained models using peptides detected from DIA data. The training parameters were set as follows: epoch, 40; warmup epoch, 10; learning rate, 0.0001; batch size, 1024. L1 loss was used for training.

### Training and testing data generation
We trained two different types of prediction models (RT and fragment ion intensity prediction) for each dataset. For each dataset, unless otherwise specified, we trained using the human DIA data and tested using the paired yeast or metaproteome DIA data. For each training or testing DIA MS data, we used DIA-NN (version 1.8.1)[26] in library-free mode to detect peptides at 1% precursor-level FDR. For the global proteome datasets, all accepted peptides were used in training and testing data generation, whereas for the phosphoDIA dataset, one additional filter was applied: PTM.Site.Confidence ≥0.75.

To prepare training data for fragment ion intensity prediction, for each precursor identified by DIA-NN that passed 1% precursor level FDR, we first extracted its matched MS2 spectrum at the apex time point from the corresponding DIA data file. We then calculated all matched fragment ions by comparing the peptide against the MS2 spectrum using a predefined mass tolerance (e.g., 10 ppm). During the matching, b and y ions were considered for global proteome data; while for phosphoproteome data, b and y ions plus phosphorylation neutral loss ion (−98 Da) were considered. Both single and double charged fragment ions were considered for precursors with charge 2+ or higher. We then used a two-pronged approach to identify fragment ions that are shared with other peptides (Fig. 1c). First, in the spectrum-centric shared peak determination step, we examined all other peptides matched to this MS2 spectrum. If a peak from the MS2 spectrum was shared by any other peptide, then the peak was labeled as "shared". An example was shown in Supplementary Fig. 1. Note that some shared peaks may not be detected using this method, if some of the peptides that generated the DIA spectrum were not identified. Therefore, we implemented a second, peptide-centric strategy for detecting shared peaks. Specifically, for each matched fragment ion from the spectrum-centric strategy, we extracted its XIC based on the peak boundaries determined by DIA-NN. The peak boundaries could be refined using a similar procedure described in a previous study[20]. For each matched fragment, we first performed Savizky-Golay smoothing[33] or weighted moving average smoothing on the fragment ion chromatograms and then calculated the Pearson correlation between the smoothed XIC and each of the smoothed chromatograms from the other matched fragment ions. We assigned an aggregate score to each matched fragment ion by summing up the Pearson correlation scores between this fragment and all other matched fragments, and we identified the top-scoring fragment ion based on this aggregate score. Finally, a peak was labeled as shared if the correlation of a fragment to the top-scoring fragment ion was less than a predefined threshold. The threshold was set to 0.8 in this study. For each peak with correlation score higher than the predefined threshold, we further calculated a peak shape score. Specifically, at each boundary time point of the peak, if the intensity of the peak at the boundary time point was higher than the median intensity of all detected peaks at the boundary time point, and the intensity was also higher than 10% (we increased the cutoff from 10% to 25% when the highest intensity of the peak was less than 50% of the highest intensity of all detected peaks) of the highest intensity of the peak between its left boundary and right boundary, we assigned a score of 1 to the peak when only one

of the two boundaries met the criteria, and a score of 2 when both boundaries met the criteria. If the peak shape score is 2, then the peak was also labeled as shared. A few examples are shown in Supplementary Figs. 2–4.

Overall, a peak was labeled as shared when it was identified as a shared peak by either of the two strategies (spectrum-centric or peptide-centric). In addition, any peak that matched to more than one peptide fragment ion (e.g., a b-ion and y-ion from the same peptide with similar m/z values) was also taken as a shared peak. Intensities of unmatched fragment ions were set to zero, unless the fragment m/z fell outside the m/z scan range, in which case the ion was masked during model training. Finally, for each annotated spectrum, the matched fragment ion intensities were linearly normalized so that the highest matched peak had a height of 1. If the highest matched peak was a shared peak or the percentage of shared peaks exceeded a specific threshold (e.g., 50%), then the spectrum was not used for training or testing.

To prepare training data for RT prediction, for a peptidoform (peptide sequence + modifications) with multiple precursors identified in different charge states, the RT of the most confident precursor was taken as the RT of the peptidoform. For a peptidoform with only one precursor identified, the RT of the precursor was taken as the RT of the peptidoform. The RTs were then normalized to the range of 0–1 by dividing the RT of each peptidoform by the maximum RT extracted from the corresponding MS/MS data or a user-defined RT value.

## Spectral library generation using Carafe

To generate an experiment-specific in silico spectral library, a single DIA run was required and was analyzed using DIA-NN. Then, a fragment ion intensity prediction model and an RT prediction model were trained using the method described above. In the model fine-tuning step, 10% of peptides were used for testing. For a given protein database, we first performed in silico protein digestion based on a specified enzyme, the maximum allowed missed cleavage sites, and peptide length limits. Next, for each digested peptide, different peptidoforms were generated based on the specified fixed and variable modifications, as well as the specified precursor charge range (by default, 2+ to 4+). The fixed modifications were set as Carbamidomethyl (C) by default. No variable modifications were applied for global proteome data while phosphorylation (STY) were set as variable modifications for phosphoproteome data by default. Finally, we applied the trained models to predict fragment ion intensities and RTs for all the peptidoforms to generate the spectral libraries. During spectral library generation, only precursors and fragment ions with m/z values within the m/z scan range of the corresponding DIA data used for model training were included in the library. Predicted RTs were converted back to the corresponding experimental RT scale by multiplying the predicted RTs by the normalization RT factor used during RT training data generation. For each precursor, the top 20 fragment ions were selected based on the predicted fragment ion intensities, and no peak masking was applied during prediction.

## Proteome sample preparation

**Human cell culture and treatment.** HeLa S3 cells were cultured at 37 °C and 5% CO2 in Dulbecco's modified Eagle's medium (DMEM) supplemented with 4.5 g/L glucose, l-glutamine, 10% fetal bovine serum (FBS), and 0.5% streptomycin/penicillin. Cells were grown to an 80% confluency. At the time of harvest, cells were left attached to plates, rinsed three times quickly with ice-cold PBS then flash-frozen in liquid nitrogen prior to storage at −80 °C.

**Yeast growth.** The S288C *S. cerevisiae* strain was selected for all downstream applications. Yeast were grown on a plate overnight, and single colonies were inoculated into media containing yeast extract peptone dextrose (YEPD). The cultures were grown to OD 0.6 before being harvested, pelleted, and frozen at −80 °C until use.

**Protein aggregation capture (PAC) sample preparation.** Protein from HeLa and yeast samples was prepared for LC-MS using a protocol modified based on magnetic bead based tryptic digestion methods reported previously[34,35]. The cells were lysed in a buffer for 5 min containing 2% SDS, 100 mM Tris buffer pH 8.5, and ThermoFisher protease inhibitors. Each sample was briefly sonicated on the Branson probe sonicator for 5 s, and total protein concentration was determined using the bicinchoninic acid (BCA) method calibrated using bovine serum albumin (Pierce, Thermo Fisher scientific)[36]. Samples were then diluted to a final concentration of 1 μg/μL. The diluted sample lysates were reduced in 20 mM dithiothreitol and alkylated in 40 mM iodoacetamide. ReSyn Hydroxyl beads were added to the reduced and alkylated samples at a ratio of 4 μL per 25 μg of protein. Protein aggregation onto the hydroxyl beads was induced by adding acetonitrile to a final concentration of 70%. The bead-bound proteins were further subjected to washes containing three 95% acetonitrile and two 70% ethanol washes. After the final wash, the samples were briefly centrifuged to remove any residual ethanol, and trypsin in 50 mM ammonium bicarbonate was added in at a ratio of 33:1 (protein to trypsin) for digestion at 47 °C for 3 h. The resulting sample peptides were eluted off the beads, dried down by a centrifuge vacuum speedvac, and frozen in the −80 °C until further use. Frozen peptide samples were resuspended to a final concentration of 500 ng/μL in 0.1% formic acid prior to mass spectrometry analysis.

## Phosphoproteome sample preparation

**Human cell culture treatment and lysis.** HeLa S3 cells were cultured at 37 °C and 5% CO2 in DMEM supplemented with 4.5 g/L glucose, l-glutamine, 10% FBS, and 0.5% streptomycin/penicillin. To generate bulk phosphopeptides for method comparisons, cells were grown to 80% confluency, incubated in serum-free medium for 6 h prior to treatment with or without 1 mM pervanadate for 15 min, followed by the addition of 10% FBS for 15 min. At the time of harvest, cells were left attached to the plates, rinsed three times quickly with ice-cold PBS then flash-frozen in liquid nitrogen prior to storage at −80 °C. Cells were harvested by scraping frozen cells from the plates in 8 M urea, 50 mM HEPES, 75 mM NaCl, pH 8.0. Cells were sonicated with six 20 s pulses at 12 W with equal rests in ice. The lysate was clarified by centrifugation at 7197 × *g* for 25 min at 20 °C. Total protein concentration was estimated using the BCA method (Pierce, ThermoFisher Scientific). Proteins were reduced with 5 mM dithiothreitol (DTT) for 30 min at 55 °C, alkylated with 15 mM iodoacetamide for 15 min at room temperature in the dark, and then quenched with 5 mM DTT for 15 min at room temperature.

**Yeast growth and lysis.** The haploid S288C derivative BY4742 *S. cerevisiae* strain was used for global phosphopeptide enrichment. Yeast cells were grown in yeast extract peptone with 2% glucose (YEPD) at 30 °C. Overnight cultures were diluted to $OD_{600}$ of 0.1 then grown to $OD_{600}$ 0.8–0.9 before the cells were harvested by washing with ice cold water and snap freezing in liquid nitrogen. Cells were lysed by bead-beating with 0.5 mm glass beads at 4 °C in urea lysis buffer (8 M urea, 50 mM HEPES pH 8.0, 75 mM NaCl). Lysates were cleared by centrifugation at 21,000 × *g* for 10 min at 4 °C, and protein concentration was determined by BCA assay (Pierce, ThermoFisher Scientific). Proteins were reduced with 5 mM dithiothreitol for 30 min at 55 °C, alkylated with 15 mM iodoacetamide for 30 min at room temperature in the dark, and the alkylation was quenched with an additional 5 mM dithiothreitol for 15 min at room temperature.

**Digest of HeLa and yeast for global phosphopeptide enrichment.** Protein lysates were diluted five-fold in 50 mM ammonium bicarbonate and digested by trypsin at a final trypsin-to-protein ratio of 1:100 by mass. Proteins were digested at 37 °C for 15 h with mixing. Digests were quenched with 0.5% TFA (pH < 2). Quenched digests were

centrifuged at $7000 \times g$ for 5 min at room temperature to remove precipitates. Peptides were desalted on Waters SEP-PAK C18 cartridges. Briefly, columns were activated by the sequential addition of 1 column volume (CV) of methanol; 3 CVs of 100% acetonitrile (ACN); 1 CV of 70% ACN, 0.25% acetic acid (AA); 1 CV of 40% ACN, 0.5% AA; 3 CV 0.1% TFA. Acidified digests were then loaded followed by reload of the flowthrough. The column was washed with 3 CV of 0.1% TFA and 1 CV of 0.5% AA. Peptides were eluted with 0.75 CV of 40% ACN, 0.5% AA followed by 0.5 CV of 70% ACN, 0.25% AA. Peptides were dried by vacuum centrifugation and stored at −20 °C until enrichment. Phosphotyrosine containing peptides were initially depleted by phosphotyrosine-specific enrichment[37]. Remaining peptides from flow throughs were desalted, dried, and stored at −20 °C for global phosphopeptide enrichment.

**Global phosphopeptide enrichment.** Desalted peptides were resuspended in 80% ACN, 0.1% TFA at 900 µL per 250 µg peptides. Precipitates were removed by centrifugation at $21,000 \times g$ for 5 min at 4 °C and peptides were added to a 96-well plate for R2-P2 as described previously[38] except with the following modifications: peptide binding was performed in a deep well plate in 900 µL and phosphopeptides were eluted in 100 µL instead of 50 µL of 2.5% ammonium hydroxide, 50% ACN followed by acidification with 60 µL of 10% formic acid, 75% ACN. The optional filtering step was performed in which eluates were passed through two layers of C8 filter material in a 200 µL pipette tip as described previously[38]. Peptides were dried by vacuum centrifugation then resuspended in 4% formic acid, 3% acetonitrile for mass spectrometry measurement.

**Metaproteomics sample preparation**
The marine microbiome sample for metaproteomic analysis was collected June 4, 2021 at 1:00 pm PDT in East Sound, WA according to methods detailed in ref. 39. The sample filter (0.22 µm 47 mm polyethersulfone) containing the bacterial fraction of the water column (0.22–1.0 µm) was processed using mechanical lysis in 100 µL 5% SDS solution followed by three subsequent rinses of the filter with 100 µL nanopure water. The resulting 400 µL whole cell solution was collected in microfuge tubes and sonicated (Branson 250 Sonifier; 20 kHz, $30 \times 10$ s on ice). Samples were then evaporated using a SpeedVac to a final concentration of 5% SDS in 100 µL. Enolase (0.16 µL 100 ng µL$^{-1}$ enolase per 1 µg protein) was added to the sample at the start of the S-trap protocol to ensure proper sample digestion. The sample (20 µg protein) was treated with benzonase (0.5 µL 250 unit µL$^{-1}$ 10 min at 95 °C), reduced with 20 mM dithiothreitol for 10 min at 60 °C and 5 min cool down to room temperature, alkylated with 40 mM iodoacetamide for 30 min in the dark, acidified to pH < 2 (1.2% aqueous phosphoric acid), and then processed on an S-trap column according to manufacturer's recommendations. Proteins were digested with Promega modified trypsin (2 µg for 1:10 ratio, 4 h 37 °C). Purified peptides were evaporated to dryness and resuspended in 2% acetonitrile (ACN), 0.1% formic acid with final concentration of 0.5 µg protein µL$^{-1}$.

**Liquid chromatography analysis**
LC separation of the peptides was done prior to data acquisition on the mass spectrometer. Peptides were separated using a 15 cm PepSep column (150 µm ID, 1.9 µm particle size) heated to 45 °C. Reversed-phase separation on a Neo Vanquish LC system was used in conjunction with measurement on a Thermo Astral and a Thermo Exploris 480. Total proteome measurements made on the Neo Vanquish with the Thermo Astral were separated by a 24 min gradient at 1.3 µL/min flow consisting of 4–6% Buffer B in 0.7 min, 6–6.5% Buffer B in 0.3 min, 6.5–40% Buffer B in 20 min, 40–55% Buffer B in 0.5 min, 55–99% Buffer B in 3.5 min for the column wash. Total proteome data acquired on the Thermo Exploris were separated on a 90 min gradient consisting of

4–6% Buffer B with 1.3 µL/min flow for 0.7 min, 6–6.5% Buffer B with 1.3 µL/min flow for 0.3 min, 6.5–5% Buffer B with 0.8 µL/min flow for 82.7 min, 50% Buffer B with 1.3 µL/min flow for 0.5 min, 50–99% Buffer B with 1.3 µL/min flow for 2.3 min, and 99% Buffer B for 3.5 min for the column wash. For phospho-enriched samples on the Exploris, the Neo Vanquish gradient was changed to 86.2 min and consisted of 1–40% Buffer B with 0.8 µL/min flow for 82.7 min, 40–99% Buffer B with 1.3 µL/min flow for 0.5 min, and 99% Buffer B with 1.3 µL/min flow for 3 min for the column wash. For total proteome samples collected on the Thermo Lumos Tribrid Orbitrap mass spectrometers, an EvoSep One LC system running the Extended 88 min method was used or an EASY-nLC 1200 system running a 90 min LC gradient was used. Specific details regarding LC conditions are provided in Supplementary Data 1.

**Data independent acquisition mass spectrometry**
Human and yeast data were acquired using DIA on Thermo Astral, Thermo Lumos Fusion Tribrid, and Thermo Exploris 480 mass spectrometers. Within a batch of runs, a chromatogram library containing 4 (on the Astral) or 6 (on the Lumos Fusion Tribrid and Exploris 480) independent injections of the samples were analyzed. On the Orbitrap Astral, each library injection consisted of a cycle of one 240,000 resolution full MS spectrum with a mass range of 375–985 m/z, custom injection time at 50 milliseconds (ms), and standard target AGC. The full MS1 spectrum was followed by a data-independent MS2 spectrum using the Astral analyzer with 1 m/z isolation window with 27% HCD collision energy, 10 ms injection time, 250% target AGC, and varying 125 m/z wide precursor mass ranges (400–525, 525–650, 650–775, 775–900 m/z) for each injection. On the Lumos Fusion Tribrid and Exploris 480, each of the 6 library injections consisted of one cycle of 30,000 resolution full MS spectrum with a mass range of 395–1005 m/z, automatic injection time, and standard target AGC. The full scan was followed by a data-independent MS2 spectrum at 30,000 resolution using the Orbitrap analyzer with 4 m/z staggered isolation window with 27% HCD collision energy, "auto" maximum injection time on the Exploris 480 and 54 ms maximum injection time on the Fusion Lumos Tribrid, 1000% target AGC, and varying 100 m/z wide precursor mass ranges (400–500, 500–600, 600–700, 700–800, 800–900, 900, 1000 m/z) for each injection.

The individual human and yeast sample runs of DIA data on the Orbitrap Astral consists of one cycle of a 240,000 resolving power spectrum on the Orbitrap with custom injection time of 50 ms and standard target AGC, and an MS2 spectrum collected subsequently using the Astral detector using 2 m/z non-staggered isolation windows with 5 ms injection time, 400–900 m/z precursor mass range, 27% HCD collision energy, and 250% target AGC. For the Exploris 480 and the Fusion Lumos Tribrid, the human and yeast sample DIA acquisition contained one cycle of 30,000 resolution with a scan range of 395–1005 m/z, auto injection time and standard AGC. The following MS2 spectrum was performed using the Orbitrap at 8 m/z staggered isolation windows with 15k resolution. The precursor MS1 spectrum was 400–1000 m/z and 27% HCD collision energy remained the same for both instruments. On the Exploris 480, the m/z range was 200–2000 with automatic injection time and target AGC of 1000%. On the Fusion Lumos Tribrid, the m/z range was 150–2000 with an injection time of 22 ms for a target AGC of 800%. Individual metaproteomic samples were only collected on the Exploris 480 using the 8 m/z staggered windows and 15k resolution DIA method. For phospho-enriched samples, the above parameters remained the same except the MS1 precursor spectrum range of the MS2 was modified to 450–1050 m/z on the Thermo Exploris 480 mass spectrometer with isolation windows of 16 m/z. For the reCID experiment, the MS2 spectrum was only performed at 8 m/z with staggered windows and 15k resolution with 27% reCID activation energy. The other parameters remained consistent for these runs on the Fusion Lumos tribrid

for human and yeast samples. More details regarding DIA methods are provided in Supplementary Data 1.

## MS data conversion
Raw MS/MS data was converted to mzML format files using MSConvert in ProteoWizard (version 3.0.24031)[40]. The vendor's peak picking was enabled for both staggered and unstaggered isolation window DIA data while demultiplexing was enabled for the staggered isolation window DIA data. The MS/MS data in mzML format were then used for downstream analysis.

## Protein databases
Protein sequences for human (UP000005640, containing 20,597 proteins) and yeast (UP000002311, containing 6060 proteins) were downloaded from UniProt (02/2024). The metaproteome database (1,050,386 proteins) was from a previous study[39].

## DIA-NN analysis
DIA-NN (version 1.8.1)[26] analysis was performed using the following parameters: fixed modification, carbamidomethyl (C); no variable modification was set except phosphorylation was set for the phosphoproteomics data; enzyme, Trypsin/P with one missed cleavage site allowed; peptide length range, 7–35; precursor charge range, 2+ to 4+. The setting of "N-term M excision" was disabled; library generation, "IDs, RT & IM profiling". For the phosphoproteomics data, the maximum number of variable modifications was set to 1 and both parameters "Mass accuracy" and "MS1 accuracy" were set to 20. The precursor FDR was set to 1%. For the TripleTOF 5600 DIA dataset, both parameters "Mass accuracy" and "MS1 accuracy" were set to 30. All other parameters were set to their default values. For single run DIA data, the "Q.Value" from the main report was used as precursor q-value for downstream analysis. For datasets with multiple runs, the "Lib.Q.Value" from the main report was used as precursor q-value for downstream analysis.

## EncyclopeDIA analysis
For EncyclopeDIA (version 2.12.30)[20,41] analysis, we first generated two in silico spectral libraries on the yeast protein database (containing 6,060 protein sequences) for each DIA dataset. Specifically, we used Oktoberfest (version 0.6.2) with Prosit models (fragment ion intensity prediction model: Prosit_2020_intensity_HCD, RT model: Prosit_2019_irt)[12,24] to generate an in silico spectral library for each DIA dataset. In the spectral library generation step, carbamidomethyl of cysteine was considered as a fixed modification, and no variable modifications were considered. Precursor charges of 2+ to 4+ were considered. The NCE parameter was set to 27. Trypsin (without proline suppression) with one missed cleavage allowed was used and only peptides with lengths between 7 and 35 amino acids were considered in Oktoberfest. We also used Carafe to generate an in silico spectral library for each dataset. The Carafe models were fined-tuned using the human sample-derived DIA data from each dataset. In the spectral library generation step, we considered carbamidomethyl of cysteine as a fixed modification, and no variable modifications were considered. Precursor charges of 2+ to 4+ were considered. Trypsin (without proline suppression) with one missed cleavage allowed was used, and only peptides with lengths between 7 and 35 amino acids were considered in Carafe. The in silico spectral libraries generated by Oktoberfest and Carafe were then converted to "dlib" format libraries using EncyclopeDIA. The "dlib" format libraries were later used for EncyclopeDIA analysis.

For each DIA dataset, we next generated a DIA-based chromatogram library by searching a set of GPF (narrow window) DIA runs against each in silico spectral library using EncyclopeDIA.

Finally, we searched the wide window DIA runs from each DIA dataset against each of the four libraries using EncyclopeDIA. In the

analysis, the V2 scoring of EncyclopeDIA was enabled except for the TripleTOF 5600 dataset. Both precursor and fragment mass tolerance parameters were set to 40 ppm for the TripleTOF 5600 dataset. The EncyclopeDIA analysis was run through the nf-skyline-dia-ms workflow (https://nf-skyline-dia-ms.readthedocs.io).

## FDR control evaluation
FDRBench (version 0.0.1, https://github.com/Noble-Lab/FDRBench)[30] was used to generate a peptide-level entrapment database and estimate FDP for FDR control evaluation. In the evaluation, protein sequences for human (UP000005640, containing 20,597 proteins) were taken as the original target protein database. The original target proteins were in silico digested into peptides using trypsin (without proline suppression) with one missed cleavage allowed. Only peptides with lengths between 7 and 35 amino acids were considered. All "I" amino acids were converted to "L." For each original target peptide, an entrapment peptide was generated using FDRBench with the C-terminal amino acid fixed. The final peptide level entrapment database was generated by combining the original target peptides and the entrapment peptides. FDP was estimated using the paired method (upper bound) and the lower bound method using FDRBench:

$$\widehat{FDP}_{upper\ bound} = \frac{N_{\mathcal{E}} + N_{\mathcal{E} \geq s > \mathcal{T}} + 2 \cdot N_{\mathcal{E} > \mathcal{T} \geq s}}{N_{\mathcal{T}} + N_{\mathcal{E}}}, \tag{1}$$

$$\widehat{FDP}_{lower\ bound} = \frac{N_{\mathcal{E}}}{N_{\mathcal{T}} + N_{\mathcal{E}}}, \tag{2}$$

where $s$ is the discovery cutoff score; $N_{\mathcal{E} \geq s > \mathcal{T}}$ denotes the number of discovered entrapment peptides (scoring $\geq s$) for which their paired original target peptides scores $<s$; and $N_{\mathcal{E} > \mathcal{T} \geq s}$ is the number of discovered entrapment peptides for which the paired original target peptides scored lower but were still also discovered. $N_{\mathcal{T}}$ and $N_{\mathcal{E}}$ denote the number of original target and entrapment discoveries, respectively.

## Quantification precision and accuracy evaluation
The LFQBench dataset was downloaded from PRIDE with accession number PXD002952[31]. Specifically, the MS/MS raw files generated using the sample set HYE110 on TripleTOF 6600 with 64-variable windows acquisition were used in the study. The raw files were converted to mzML files using MSConvert with peak picking enabled in ProteoWizard (version 3.0.24031). A total of six MS runs were used in the study: three replicates for sample A and three replicates for sample B. In the dataset, peptides from three species (human, yeast and E.coli) were mixed together to generate samples A and B with expected A:B ratios of 1:1 for human, 10:1 for yeast and 1:10 for E.coli peptides. The first MS run (labeled as 001) from the dataset was used as the training MS run for Carafe. Five spectral libraries were generated. Four of them were generated using Carafe, and one of them was generated using DIA-NN's built-in model. The parameters for library generation were the same as described in the previous section. The MS/MS data were searched against each spectral library using DIA-NN (version 1.8.1) with the following parameters: fixed modification, carbamidomethyl (C); no variable modification was set; enzyme, Trypsin/P with one missed cleavage site allowed; peptide length range, 7–35; precursor charge range, 2+ to 4+. The precursor FDR was set to 10%. The DIA-NN analysis was run through the nf-skyline-dia-ms workflow (https://nf-skyline-dia-ms.readthedocs.io). The column "Precursor.Normalised" from the main report generated by DIA-NN was used as precursor intensity for the quantification precision and accuracy evaluation.

## False localization rate evaluation
The yeast phosphoproteome DIA dataset used for false localization rate evaluation was downloaded from MassIVE with accession number

MSV000093613[32]. The MS/MS data were generated on a Thermo Astral instrument using a phosphopeptide enriched yeast sample with 225 synthetic spike-in phosphopeptide standards. In the current study, only the synthetic phosphopeptide standards with a single phosphorylated site (205 phosphopeptides) were considered. A single human phosphoproteomics DIA MS run generated from the same study with a similar instrument setting was downloaded and used for Carafe model fine-tuning. The MS/MS raw files were converted to mzML files using MSConvert with peak picking enabled in ProteoWizard (version 3.0.24031). The human DIA data was analyzed using DIA-NN and used to fine-tune Carafe models. In total, five spectral libraries were generated. Four were generated through Carafe, and one was generated using DIA-NN's built-in model. Each of the libraries contained the synthetic phosphopeptide standards and yeast peptides. Each yeast phosphoproteome DIA MS run was searched against each of the spectral libraries using DIA-NN (version 1.8.1) with the following parameters: fixed modification, carbamidomethyl (C); variable modification, phosphorylation (STY); a maximum of one variable modification allowed; enzyme, Trypsin/P with one missed cleavage site allowed; peptide length range, 7–35; precursor charge range, 2+ to 4+; precursor and fragment ion tolerance, 10 ppm. The precursor FDR was set to 1%. Two additional filters were applied: PTM.Q.Value ≤ 0.01, and PTM.Site.Confidence ≥0.75.

## EGF-stimulated HeLa phosphoproteome DIA data analysis

The EGF-stimulated HeLa phosphoproteome DIA dataset was generated on a Thermo Astral instrument in a previous study[32]. A total of six MS/MS raw files were downloaded from MassIVE with accession number MSV000093613. The dataset includes two sample groups (a control group and an EGF stimulated group), and each group has three replicates. The MS/MS raw files were converted to mzML files using MSConvert with peak picking enabled in ProteoWizard (version 3.0.24031). A single human phosphoproteomics DIA MS run was used for Carafe model fine-tuning. In total, five spectral libraries were generated. Four were generated through Carafe, and one was generated using DIA-NN's built-in model. The MS/MS data were searched against each of the spectral libraries using DIA-NN (version 1.8.1) with the following parameters: fixed modification, carbamidomethyl (C); variable modification, phosphorylation (STY); a maximum of one variable modification allowed; enzyme, Trypsin/P with one missed cleavage site allowed; peptide length range, 7–35; precursor charge range, 2+ to 4+; precursor and fragment ion tolerance, 10 ppm. MBR was enabled. The precursor FDR was set to 1%. Two additional filters were applied: PTM.Q.Value ≤ 0.01, and PTM.Site.Confidence ≥ 0.75. Only peptidoforms that were detected in at least two replicates in each of the two groups or in all three replicates of one group were used for downstream quantitative analysis. For differential expression analysis, for peptidoforms detected in all three replicates in at least one of the two groups, if that peptidoform was not detected in the other group or was only detected in one of the replicates in the other group, then the quantification values in each of the missing replicate were imputed using the function "impute.knn" from the R package impute (version 1.76.0). Fold changes of peptidoforms were calculated as the ratio of the median quantification values of the peptidoforms in the EGF stimulated group to the median quantification values of the peptidoforms in the control group. The fold changes were log2 transformed. The p-values were calculated using a two-tailed t-test on the log2 transformed quantification values. Following the threshold used in the original study[32], peptidoforms with $p ≤ 0.05$ and log2 transformed fold changes ≥1 or ≤−1 were considered as significantly regulated.

## Spectral library generation using NCE calibration

For NCE calibration for a given dataset, we first predicted the fragment ion intensities for each peptide in the training dataset used by Carafe using the AlphaPeptDeep pretrained model with a range of NCE values (between 20 and 40). The similarity between the predicted fragment ion intensities and the observed fragment ion intensities was estimated using the unweighted spectral entropy similarity metric[28,42]. Only fragments that were determined as unshared by Carafe were used. For every NCE value, we computed the median similarity score, and the NCE value that yielded the highest median was chosen as optimal. Finally, the spectral library was generated using the AlphaPeptDeep model configured with this optimal NCE setting.

### Reporting summary

Further information on research design is available in the Nature Portfolio Reporting Summary linked to this article.

## Data availability

The MS/MS datasets generated in this study have been deposited to Panorama Public (ProteomeXchange identifier: PXD056793) and are available at https://panoramaweb.org/Carafe.url. The Q Exactive HF-X dataset and the TripleTOF 5600 dataset were downloaded from PRIDE[43] with the accession number PXD028735[44]. The LFQBench dataset was downloaded from PRIDE with accession number PXD002952[31]. Both the yeast phosphoproteome DIA dataset with spike-in synthetic phosphopeptides and the EGF-stimulated HeLa phosphoproteome DIA dataset were downloaded from MassIVE with the accession number MSV000093613[32]. The MS/MS files used in this study are listed in Supplementary Data 2. Source data are provided with this paper.

## Code availability

Carafe is available under the Apache 2.0 license at https://github.com/Noble-Lab/Carafe. A Nextflow workflow for Carafe is available under the Apache 2.0 license at https://nf-carafe-ai-ms.readthedocs.io/. The Skyline version with Carafe integrated is available at https://skyline.ms/carafe.url. The source code of the customized AlphaPeptDeep used in this study, with peak masking support, is available under the Apache 2.0 license at https://github.com/wenbostar/alphapeptdeep_dia.

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

## Acknowledgements

This work was supported by National Science Foundation award 2245300, National Institutes of Health awards R24 GM141156 and U19 AG065156, NIH/NIGMS grant R35GM152061 (J.V.), the National Science Foundation Graduate Research Fellowship Program (Grant No. DGE-2140004, B.W.), and by the Intelligence Advanced Research Projects Activity (IARPA) TEI-REX program through the Army Research Office contract W911NF2220059. We gratefully acknowledge Gennifer Merrihew for testing Carafe.

## Author contributions

Conceptualization: B.W., M.J.M., W.S.N. Data curation: B.W., C.H., M.J.M. Formal analysis: B.W., M.J.M., W.S.N. Funding acquisition: B.W., M.J.M., W.S.N. Investigation: B.W., M.J.M., W.S.N. Methodology: B.W., M.J.M., W.S.N. Software: B.W., D.S., W.-F.Z., M.R., B.X.M. Resources: C.H., M.J.M., W.S.N. Supervision: B.W., M.J.M., W.S.N. Visualization: BW. Writing—original draft: B.W., M.J.M., W.S.N. Writing—review and editing: B.W., C.H., D.S., W.-F.Z., M.R., A.C., M.M., B.L.N., B.X.M., M.D.B., J.V., M.J.M., W.S.N.

## Competing interests

The MacCoss Lab at the University of Washington receives funding from Agilent, Bruker, Shimadzu, Thermo Fisher Scientific, and Waters to

support the development of Skyline, a quantitative analysis software tool. MJM is a paid consultant for Thermo Fisher Scientific. The remaining authors declare no competing interests.
