## [Transparent Peer Review file · Nature Communications]

Carafe enables high quality in silico spectral library generation for data-independent acquisition proteomics

Corresponding Author: Dr William Noble

Version 0:

Reviewer comments:

Reviewer #1

(Remarks to the Author)

The manuscript by Wen et al. introduces Carafe, a novel tool for generating high-quality experiment-specific in silico spectral libraries for data-independent acquisition (DIA) proteomics using fine-tuning. The authors demonstrate that Carafe improves fragment ion intensity and retention time prediction resulting in improved peptide detection compared to existing pre-trained models. A key aspect of Carafe is its ability to detect and mask interfered fragment ions for model training, addressing the described challenge of chimeric spectra in DIA data. While the manuscript is concise and well-organized, several major and minor points need to be addressed:

Major Points:

- With the information provided, I fail to see the substantial novelty in the approach. The bottom line of the paper appears to be: a) fine-tuning removes biases from a pre-trained model, b) RT fine-tuning is required to yield high-quality predictions since pre-trained models do not generalize well over different setups, and c) do not expect good performance when using pre-trained models for different mass analyzers. Fine-tuning has been proposed and used frequently over the last few years. While it is nice to bundle this up in a reusable package (although it seems to be limited to users with some level of bioinformatics experience), this can only be viewed as an incremental improvement over state-of-the-art since the authors of AlphaPeptDeep already suggested and used fine-tuning in the original manuscript. I am not surprised at all that fine-tuned models improve peptide detection, particularly in cases where non-matching models are used. Whether or not the masking is required is not clear at all. My assumption is that masking provides a minor benefit over fine-tuning models without masking.
- I am surprised by the shape and effect of the fine-tuning (Figure 2 and 4) on spectral similarity. First, I would have expected a larger difference in spectral similarity before and after fine-tuning, for a) precursors of higher charge (e.g. visible in Figure 2 by at least a second distribution of points from i.e. 3+ precursors that benefit from fine-tuning more than others i.e. 2+ precursors) and b) spectra that are not predicted well by the pre-trained model. As of right now, fine-tuning seems to have a very similar effect on all spectra, irrespective of prior performance (very low similarity gain the same during fine-tuning as very high similarity cases; guesstimated linear regression has a slope of 1). Second, the distribution and percent spectra above the diagonal do not correlate with the reported gains. While it is impossible to assign a case in Figure 2 to a case in Figure 4 due to limited Figure description, the gains on e.g. Exploris data are not in line with the overall distribution of spectral similarities (judged by the lack of outliers). Third, the reported negative correlations before and after fine-tuning make me wonder if the models are learning anything substantial at all. All this seems to indicate that the fine-tuning is not doing what the authors argue ("This difference arises largely due to differences in collision energy optimization between DDA and DIA").
- There is no analysis presented to demonstrate that the quantitative values associated with newly identified peptides align with biological expectations. This omission makes it difficult to assess the validity and relevance of the additional identifications gained through Carafe.
- In Figure 4, the authors do not provide sufficient evidence to demonstrate that the novel intensity prediction approach is the primary factor driving performance improvements. Visualization suggests that retention time (RT) prediction has a significant impact, particularly when the original model was trained on similar machines or under the same experimental conditions. To substantiate the claim that the peak masking strategy and new loss function are crucial, the authors should conduct a comparative analysis between models trained with and without masking. This comparison would help differentiate between the effects of transfer learning and the proposed peak masking approach, thereby clarifying the true source of the observed performance gains.

- The authors do not provide experimental validation of false discovery rate (FDR) control, which is crucial given that the model is trained on data filtered at 1% FDR from the same experiment. This raises concerns about potential data leakage and the reliability of the reported results. The authors should demonstrate whether this new model introduces a bias in the scores of targets over decoys. From the text, it is also not clear if the generated library for data analysis contains all peaks, or if the predictions are also masked.

Minor Points:

- Please elaborate on the details of the fine-tuning, particularly the data generation.
- No evaluation/analysis is provided that would allow the assessment of possible overfitting. The authors do not provide a detailed analysis of the amount of data required for fine-tuning. Given that the authors use a fixed number of epochs for fine-tuning, I would expect the data-sets size to be important.
- I assume the pre-trained AlphaPeptDeep model supports different (N)CE settings. Please provide evidence that a much less resource-intensive calibration of the NCE does not yield similar results for e.g. the Exploris data – for each precursor charge and possibly offset to the center of the isolation window do account for differences in the way how NCE is calculated between DDA and DIA.
- In Figure 2, “TripleTOF 5600, SWATH” is listed 3 times, with varying improvements as a result of fine-tuning. However, the “n” is the same, which I find unlikely to happen if different datasets are plotted.
- In Figure 2, the axis labels are unclear and do not specify what is being plotted. This information is only provided in the legend. The authors should add proper axis labels for clarity.
- The plotting of Figure 2 could be improved by using a 2D density. The current visualization does not highlight the majority of the data points.
- For the peptide-centric and spectrum-centric masking approaches, the authors should provide information on the average number of peaks masked compared to the original number of peaks. This would help assess whether the majority of peaks are masked and how this affects the selection of correct identifications.
- In Figure 4, the authors present the numbers of identifications but do not illustrate the overlap between different methods. It is important to demonstrate whether the identifications made with the new retrained models are shared with other methods or are entirely novel.

(Remarks on code availability)

No documentation on source-code level. Github only contains the "first release", no history of the developed code, implying that the code was/is developed somewhere else. Most (if not all) logic is in AIGear.java, which does not follow good coding practices. Not a fan of the duplication of the AlphaPept code base, which requires the authors to keep their copy up to date as well now.

Reviewer #2

(Remarks to the Author)

In this study, Bo Wen et al. developed Carafe, a tool to predict high-quality spectral libraries for DIA analysis. Unlike conventional approaches relying on DDA data, this tool used DIA data to fine-tune a pre-trained model (provided by AlphaPeptDeep, in this case). This approach was shown to generate better spectral libraries, featured by more accurate fragment ion intensities and retention times. The concept of integrating information extracted directly from DIA data into spectral library generation is undoubtedly appealing. It has the potential to improve the re-analysis of DIA data itself or the data from similar MS settings. Furthermore, building an interference-free spectral library is essential to avoid misleading peptide quantification caused by shared fragment ions. Overall, I think the proteomics community would benefit from this tool and its underlying idea. To further improve the quality of the manuscript, I wish the authors may consider to address the following points:

1. Most results in the paper were about qualitative analysis -- carafe was shown to outperform other methods in terms of the number of precursors identified. DIA-MS is particularly valued for its strength in quantitative proteomics. The manuscript could benefit from comparative analysis using quantitative metrics such as quantitative accuracy, relative errors, and coefficients of variation (CV) among replicates. Several DIA benchmarking datasets, such as those in PMID: 36609502 and PMID: 27701404 have been generated for such purpose by mixing proteins with known concentrations and could be utilized to benchmark the tool's performance quantitatively. Demonstrating improvements in these metrics would provide stronger evidence of Carafe's utility.
2. One significant contribution of carafe is its ability to detect interference peaks arising from shared fragment ions. However, the manuscript does not clearly demonstrate the downstream benefits of removing such misleading signals. The authors could leverage gold-standard DIA datasets to illustrate these advantages. For example: How much do relative errors decrease when interference is removed? How do different strategies for determining shared peaks compare in terms of their impact on accuracy?
3. For the phosphoproteome analysis, the authors may consider validating Carafe using synthetic phosphoproteome datasets, particularly focusing on phosphosite localization accuracy. Ideally, an interference-free library or one built with discriminative fragment ions is expected to help with PTM site localization and quantification of peptides with PTMs. Including such analyses would broaden the tool's applicability, especially for PTM data.
4. ROC curves and comparisons at the protein level may be considered to be added.
5. The authors may consider giving more explanation or illustration. For example, figure 1b and 1c seem not clear enough

(e.g. peaks across DIA and DDA spectra were not aligned; the numbers in the tables seem to be random and are difficult to interpret...). The authors are encouraged to refine these figures to make them more intuitive and easier to follow. More real-world examples (as in the format of Figure 1c) could be added to explicitly showcase the advantages of removing shared peaks using complementary strategies.

6. Figures 2 and 3 highlight Carafe's improvements. However, the manuscript would benefit from including additional concrete results, such as a few gained peptides with corresponding predicted spectra and real spectra with annotated peaks.

7. While the manuscript is understandable, the paper, if possible, can be edited again by a native English speaker.

(Remarks on code availability)

Version 1:

Reviewer comments:

Reviewer #1

(Remarks to the Author)

I commend the authors for the work they put into the response and am happy to see most points addressed. I am particularly impressed by the integration in Skyline which boost applicability and value of the proposed approach and the new data and analysis added. However, two major and two minor points remain to be addressed.

1) On the importance of masking, the authors show that a 2% increase can be achieved when fine-tuning a model with masking compared to fine-tuning without masking. This is a rather marginal improvement and appears to support my point that masking is likely not as important as the authors make it appear in the paper (2 Figures + a large body of text). I would like to see a more systematic evaluation of this in the manuscript and depending on the results an adjustment of wording (wrt novelty and importance of this) by doing the analysis the authors did at least 3 times (technical replicates) for masking and no masking for more than just one example (as best covering different MS instruments that are "close" or "far away" from the original model). The example the authors choose could be an outlier since TOF data tends to be more noisy compared to Orbitraps and masking may e.g. largely remove noise from the training. At this point, it is not clear whether the 2% gains are likely to be observed for other MS instruments and to which degree the 2% gains are a result of (partially) non-deterministic algorithms (both the fine-tuning and I assume DIA-NN are not deterministic). Judging by eye, Figures R6 and R7 show that there is quite a number of unique precursors confidently identified by each of the 5 evaluated methods, that may as well represent the 2%. I understand the reasoning that masking may potentially be important when attempting to fine-tune models based on more complex DIA data, however, as far as I can tell right now, this is not supported with evidence, yet. Such a claim may be substantiated if the increase observed when comparing masked vs non-masked fine-tuning correlates with the "complexity" of the DIA data it was trained on, as stated by the authors by the discussion.

2) I am aware that "peptide detection in DIA workflows typically involves multiple complex steps, and the search tool that we used in this analysis is not open source". However, I politely disagree with the consequence that one should not attempt to understand why an increase in SPC of only 0.05 leads to substantial gains in identified precursors. To me, and I would assume most readers as well, the reported improvement in prediction accuracy appears very small in relation to the reported gains. I suggest that the authors attempt to investigate the relation between increased SPC and observed gains. For example, the authors could check whether the precursors gained are cases of formerly low but now high SPC (after fine-tuning), whether those show a substantially bigger improvement compared to other precursors (e.g. the shared or lost). Are the lost peptides largely the ones which have lower SPC after fine-tuning? Do the gained peptides show a similar increase in SPC as the peptides in the test-set? Related to this is whether the authors would expect to see another increase in IDs if a different fine tuning method would result in an additional boost in performance of on average 0.05 SPC?

Minor:

1) While the authors show some example spectra of improved performance of the fine-tuned model (Figures S11 and S12), the examples picked appear to be (without exception) extreme examples for improved performance. The average, as depicted in Figure S6, is much lower than the (estimated) average 0.4 improvements shown. In my view, this leaves an incorrect impression about the performance of fine-tuning and the authors should show examples of at least average performance as well.

2) The CE calibration plots added (Figure S14) show an unexpected characteristic. One would have expected a more substantial difference between e.g. NCE 20, 29, and 40. However, the spectral entropy shows almost no difference along NCE. I encourage the authors to double check if the e.g. input to the model was provided correctly. In other examples one can find online, the difference in prediction performance is reported to be much higher when choosing the incorrect NCE. This may suggest that the pre-trained model is not as aware of the impact of NCE as reported by other tools and thus the results reported here are not representative of what NCE calibration may achieve. This may warrant a brief mention in the discussion.

(Remarks on code availability)

Despite the authors stating to improve the code for the next release, which is now in release 1.0.0, no changes were made. Most (if not all) logic is in AIGear.java. No source code documentation provided.

Reviewer #2

(Remarks to the Author)

Overall, I think the authors did excellent work addressing all of my comments, and the manuscript can now be accepted for publication.

Some suggestions regarding writing and formatting:

1. In Figure S1 of the supplementary materials, "The top panel shows..." should be revised to "The bottom panel shows the predicted fragment ion intensity values using the pretrained AlphaPeptDeep model."
2. In Figure 5, it seems that "Panel g was generated using BioRender"
3. In Figure 7, the phrase "Precursor level" appears consistently across panels a-e. As no comparison was performed at the peptide/protein level, this phrase is redundant and can be removed.

Indeed, some clarification on comment #4: My previous suggestion about "ROC curves and comparisons at the protein level" specifically referred to counting the numbers of proteins, not controlling the FDR at the protein level.

4. References 26 and 41 appear to be duplicates; please check.

(Remarks on code availability)

Version 2:

Reviewer comments:

Reviewer #1

(Remarks to the Author)

The authors have addressed all issues nicely. The manuscript can now be accepted for publication.

(Remarks on code availability)

I am happy to see more documentation in AIGear.java, however, I believe most of the rest of the code would also benefit from documentation. This should not delay publication, but I recommend checking "Read the docs" or similar features in GitHub.

Reviewer #1

The manuscript by Wen et al. introduces Carafe, a novel tool for generating high-quality experiment-specific in silico spectral libraries for data-independent acquisition (DIA) proteomics using fine-tuning. The authors demonstrate that Carafe improves fragment ion intensity and retention time prediction resulting in improved peptide detection compared to existing pre-trained models. A key aspect of Carafe is its ability to detect and mask interfered fragment ions for model training, addressing the described challenge of chimeric spectra in DIA data. While the manuscript is concise and well-organized, several major and minor points need to be addressed:

We thank the reviewer for this accurate summary of our contributions.

Major Points:

1. With the information provided, I fail to see the substantial novelty in the approach. The bottom line of the paper appears to be: a) fine-tuning removes biases from a pre-trained model, b) RT fine-tuning is required to yield high-quality predictions since pre-trained models do not generalize well over different setups, and c) do not expect good performance when using pre-trained models for different mass analyzers. Fine-tuning has been proposed and used frequently over the last few years. While it is nice to bundle this up in a reusable package (although it seems to be limited to users with some level of bioinformatics experience), this can only be viewed as an incremental improvement over state-of-the-art since the authors of AlphaPeptDeep already suggested and used fine-tuning in the original manuscript. I am not surprised at all that fine-tuned models improve peptide detection, particularly in cases where non-matching models are used. Whether or not the masking is required is not clear at all. My assumption is that masking provides a minor benefit over fine-tuning models without masking.

We agree with the reviewer that fine-tuning has been proposed and shown to be useful in analyzing DDA data in previous studies. The major methodological novelty of our work is that we propose a framework to generate high-quality spectral libraries tailored to a specific DIA experiment by fine-tuning both retention time and fragment ion intensity prediction models directly on DIA data rather than using DDA data. The challenge of using DIA data for training a fragment ion intensity prediction model is that the DIA spectra are often chimeric and contain many interfered peaks. We propose a two-pronged approach to accurately detect DIA fragment ion peaks that suffer from interference and we propose a peak masking strategy to remove the interfered peaks from the training data to improve the performance of fragment ion intensity prediction. While revising our manuscript, we have performed a new analysis (described in detail below) to show that more peptide precursors were identified using Carafe with peak masking than Carafe without peak masking. We agree with the reviewer that masking is not as significant as fine-tuning retention time prediction, but we think it is useful since the degree of interference in DIA spectra is often unknown and varies across experiments and, accordingly, the effect of training with interfered peaks is difficult to predict.

To make Carafe widely accessible to the proteomics community, as part of our revisions we have implemented Carafe in Skyline. Hence, users can now use Carafe through an intuitive graphical user interface. All of Carafe's dependencies are automatically installed by Skyline when Carafe is first launched in Skyline. A preview build of Skyline with Carafe supported is provided at <https://proteome.gs.washington.edu/~dshteyn/SkylineCarafePreview/>.

2. I am surprised by the shape and effect of the fine-tuning (Figure 2 and 4) on spectral similarity. First, I would have expected a larger difference in spectral similarity before and after fine-tuning, for a) precursors of higher charge (e.g. visible in Figure 2 by at least a second distribution of points from i.e. 3+ precursors that benefit from fine-tuning more than others i.e. 2+ precursors) and b) spectra that are not predicted well by the pre-trained model. As of right now, fine-tuning seems to have a very similar effect on all spectra,

irrespective of prior performance (very low similarity gain the same during fine-tuning as very high similarity cases; guesstimated linear regression has a slope of 1). Second, the distribution and percent spectra above the diagonal do not correlate with the reported gains. While it is impossible to assign a case in Figure 2 to a case in Figure 4 due to limited Figure description, the gains on e.g. Exploris data are not in line with the overall distribution of spectral similarities (judged by the lack of outliers). Third, the reported negative correlations before and after fine-tuning make me wonder if the models are learning anything substantial at all. All this seems to indicate that the fine-tuning is not doing what the authors argue (“This difference arises largely due to differences in collision energy optimization between DDA and DIA”).

In response to these points, we have performed an analysis to compare the performance of fragment ion intensity prediction across different precursor charge states using the pretrained DDA model from AlphaPeptDeep and the Carafe fine-tuned model on the TripleTOF 5600 dataset. As shown in Figure R1, we observed that precursors with higher charge states tend to benefit more from fine-tuning. This is consistent with the reviewer’s expectation. We have included this analysis in the revised manuscript in the section “Carafe yields accurate fragment ion intensity and RT predictions for diverse DIA datasets” as shown below:

We observed that precursors with higher charge states tend to benefit more from fine-tuning (Supplementary Figure R1).

Figure R1: **Fragment ion intensity prediction improvement across precursor charge states.** Spearman correlation coefficients between predicted and observed fragment ion intensities were calculated for each peptide precursor using both the DIA fine-tuned model and the pretrained DDA model on the TripleTOF 5600 dataset. The y-axis shows the difference in Spearman correlation (fine-tuned minus pretrained) for each precursor charge state. The numbers in the first row in the plot are the number of peptide precursors in each precursor charge state group used for the analysis. The numbers in the second row are the median Spearman correlation difference values for each precursor charge state group. In the boxplot, centerline indicates the median, the lower and upper hinges correspond to the first and third quartiles, whiskers indicate the 1.5 interquartile range and points indicate outliers.

On the other hand, peptide detection in DIA workflows typically involves multiple complex steps, and the search tool that we used in this analysis is not open source. Consequently, it is difficult to directly correlate improvements in fragment ion intensity prediction with gains in peptide detection across different datasets. We observed substantially larger increases in peptide detection on the Exploris metaproteomics DIA dataset than on other datasets using similar fragmentation methods; this discrepancy may reflect the greater impact of improved predictions in a larger search space. Indeed, our metaproteome database contained 1,050,386 protein sequences—vastly exceeding the 6,060 proteins in the yeast database used for the other analyses. Consequently, we believe that applying fine-tuning to DIA analyses with extensive search spaces, such as metaproteomics and immunopeptidomics, merits further investigation.

3. There is no analysis presented to demonstrate that the quantitative values associated with newly identified peptides align with biological expectations. This omission makes it difficult to assess the validity and relevance of the additional identifications gained through Carafe.

To address this question, we have performed a new analysis to demonstrate that the quantitative values associated with newly identified peptides align with biological expectation. We have added the result to a new section, “Quantification evaluation” in the revised manuscript as shown below:

To further assess the quantification performance of the Carafe fine-tuned library, we quantitatively compared it with the DIA-NN built-in library using an EGF-stimulated HeLa phosphoproteome DIA dataset from a previous study [1]. We first compared the phosphopeptide precursors detected using different libraries. The fully trained Carafe library outperforms all other libraries (Figure R2f), which is consistent with our previous analysis in Figure 4c. We then focused on the differentially expressed phosphopeptides identified using the Carafe library and the DIA-NN library. As shown in Figure R2g, the Carafe fine-tuned library identified 20% more differentially expressed phosphopeptides than the DIA-NN library. Notably, as shown in Figure R2h, the Carafe library detected 14 EGFR phosphosites, compared with only 8 identified by the DIA-NN library. Among the 14 EGFR phosphosites detected with Carafe, 7 were differentially expressed between the EGF-stimulated and control samples, whereas only 4 of the 8 EGFR phosphosites identified using the DIA-NN library were differentially expressed. Importantly, all three of the uniquely detected differentially expressed phosphosites were also reported in the original study [1]. Given that EGF-stimulated cells are known to exhibit a marked increase in EGFR phosphorylation, these results demonstrate that the quantitative measurements of the newly identified EGFR phosphopeptides using the Carafe library are in agreement with biological expectation.

Figure R2: **Quantification evaluation of Carafe with DIA-NN.** Quantitative evaluation using the LFQbench DIA dataset: (a)–(e). (a) Comparing precursors detected using five different *in silico* spectral libraries on the LFQbench dataset. (b) The distribution of CVs of precursor intensity on the LFQbench dataset. Numbers in the first row of the boxplots are the numbers of precursors quantified. Numbers in the second row are the median CVs. (c–e) Log-transformed ratios ($\log_2(A/B)$) of precursors over log-transformed intensities of sample B for the DDA, the DIA-NN and the Carafe fine-tuned (MS2/RT) spectral libraries on the LFQbench dataset. Dashed black lines indicate the expected $\log_2(A/B)$ ratios: 1:1 for human, 10:1 for yeast and 1:10 for *E.coli*. The number under each line is the number of precursors quantified from a species used in the plot. The boxplot on the right side of each plot shows the distribution of the $\log_2(A/B)$ ratios. Quantitative evaluation using an EGF-stimulated HeLa phosphoproteome DIA dataset: (f)–(h). (f) Comparing phosphopeptide precursors detected using five different *in silico* spectral libraries. (g) Comparing differentially expressed phosphopeptide precursors detected using Carafe fully fine-tuned and DIA-NN libraries. (h) Volcano plots of phosphosites between EGF stimulation and control. EGFR phosphosites were labeled. The differentially expressed EGFR phosphosites were highlighted in red. For boxplots, centerline indicates the median, the lower and upper hinges correspond to the first and third quartiles, whiskers indicate the 1.5 interquartile range and points indicate outliers.

4. In Figure 4, the authors do not provide sufficient evidence to demonstrate that the novel intensity prediction approach is the primary factor driving performance improvements. Visualization suggests that retention time (RT) prediction has a significant impact, particularly when the original model was trained on similar machines or under the same experimental conditions. To substantiate the claim that the peak masking strategy and new loss function are crucial, the authors should conduct a comparative analysis between models trained with and without masking. This comparison would help differentiate between the effects of transfer learning and the proposed peak masking approach, thereby clarifying the true source of the observed performance gains.

As suggested, we have performed a new analysis to compare the performance of models trained with and without masking on the TripleTOF 5600 dataset. Using the library generated without peak masking, 35,134 precursors were detected while 35,837 precursors were detected using the library generated with peak masking. The results show that the model trained with masking outperforms the model trained without masking, indicating that the peak masking strategy is useful for improving the performance of peptide detection with DIA-NN search. We agree with the reviewer that the effect of the masking strategy is not as large as the effect of fine-tuning retention time prediction, but we believe that the peak masking strategy is still useful because the degree of interference in DIA spectra is often unknown and varies across experiments. Therefore, we think it is useful to include the peak masking strategy when fine-tuning on DIA data. We have added the new analysis to the revised manuscript in the section “Carafe increases the number of peptides detected with DIA-NN” as shown below:

To further investigate the utility of fine-tuning of fragment ion intensity predictions, we generated a spectral library using Carafe without peak masking on the TripleTOF 5600 dataset. A total of 35,134 precursors were detected using DIA-NN with this library, which is 2% lower than the number of precursors (35,837) detected with the library generated using Carafe with peak masking.

5. The authors do not provide experimental validation of false discovery rate (FDR) control, which is crucial given that the model is trained on data filtered at 1% FDR from the same experiment. This raises concerns about potential data leakage and the reliability of the reported results. The authors should demonstrate whether this new model introduces a bias in the scores of targets over decoys. From the text, it is also not clear if the generated library for data analysis contains all peaks, or if the predictions are also masked.

In the study, we trained the models using human DIA data but tested the models on yeast or metaproteomics data DIA. Therefore, we think data leakage is likely not an issue in our analysis, since the peptides in the training species and the test species are very different. To demonstrate the reliability of the reported results, we have now performed a new analysis to evaluate the FDR control using our previously described entrapment strategy [2]. We observed that the FDR control is similar between using spectral libraries derived from the DDA-trained models and using spectral libraries derived from Carafe fine-tuned models (Figure R3a–c). We have added a paragraph to describe the new analysis in the section “Carafe increases the number of peptides detected with DIA-NN”:

To further investigate the quality of the spectral libraries generated by Carafe, we evaluated the FDR control at the precursor level using DIA-NN with different spectral libraries on the TripleTOF 5600 human DIA dataset (Figure R3a). We did this employing an entrapment strategy, using two different techniques that yield an upper bound and a lower bound, as previously described [2]. We observed that the FDR control is similar between the DDA-trained models

(Figure R3b–c) and Carafe fine-tuned models (Figure R3d–f). In particular, when we analyzed the same human DIA data and compared the estimated false discovery proportion (FDP) using the upper bound method for DIA-NN with three different spectral libraries at 1% FDR threshold reported by DIA-NN, we obtained estimates of 1.07% using the spectral library generated by the pretrained DDA models (Figure R3b), 1.00% using the spectral library generated by DIA-NN (Figure R3c), and 1.09% using a Carafe fine-tuned spectral library trained on a yeast DIA run from the same experiment (Figure R3d). Furthermore, the estimated FDP was 1.07% using a Carafe fine-tuned spectral library trained on a replicate DIA run (Figure R3e), while the estimated FDP was 1.02% using a Carafe fine-tuned spectral library trained on the same DIA run (Figure R3f). These results suggest that neither data leakage nor overfitting is an issue in Carafe library generation with fine-tuned models.

Figure R3: **FDR control and false localization rate evaluation.** FDR control evaluation was performed using an entrapment strategy on the TripleTOF 5600 human DIA dataset using five different *in silico* spectral libraries (a): (b) a spectral library generated using the pretrained DDA models from AlphaPept-Deep; (c) a spectral library generated using DIA-NN’s built-in model; (d) a spectral library generated using Carafe with models fine-tuned on a yeast DIA run from the same experiment; (e) a spectral library generated using Carafe with models fine-tuned on a replicate human DIA run from the same experiment, and (f) a spectral library generated using Carafe with models fine-tuned on the same human DIA run. The x-axis shows different FDR thresholds reported by DIA-NN while the y-axis shows the estimated FDPs using the upper bound and lower bound methods. The dashed vertical line is at the 1% FDR threshold, as are the numbers reported in the text in the figure. False localization rate evaluation was performed using a yeast phosphoproteome DIA dataset generated with a set of spiked-in, synthetic phosphopeptides (g). A human phosphoproteomics DIA MS run generated in the same study was used as training data to generate fine-tuned spectral libraries by Carafe. The data was analyzed using DIA-NN with different spectral libraries: (h) phosphopeptide precursors detected using different spectral libraries; (i) false localization rate evaluation using synthetic phosphopeptides detected with different spectral libraries. Panel f was generated using BioRender.

The generated libraries for data analysis contain the top 20 peaks for each precursor. There is no masking during prediction since the masking is only used during model training, and the model is not trained to predict which peaks are likely to be interfered for a given peptide. We have added a sentence to the methods section to clearly state that the predictions are not masked:

For each precursor, the top 20 fragment ions were selected based on the predicted fragment ion intensities, and no peak masking was applied during prediction.

Minor Points:

1. Please elaborate on the details of the fine-tuning, particularly the data generation.

We have added more details on the fine-tuning and data generation procedures, as highlighted in the methods sections of “Deep learning models” and “Training and testing data generation”.

2. No evaluation/analysis is provided that would allow the assessment of possible overfitting. The authors do not provide a detailed analysis of the amount of data required for fine-tuning. Given that the authors use a fixed number of epochs for fine-tuning, I would expect the data-sets size to be important.

In the present study, we mainly use DIA data from human samples to train the models and then test the models on yeast or metaproteome DIA data. Because the sets of peptides in the training and test sets are largely disjoint, we believe that the results in the manuscript are unlikely to be impacted by overfitting. During the revision of the manuscript, we have evaluated FDR control in peptide detection using a variety of spectral libraries. The presence of overfitting might introduce a bias, allowing the model to distinguish targets from decoys and thus affecting FDR control. However, as shown in the new Figure 5a-f (described in more detail in the response to a previous comment above), we do not observe obvious differences in FDR control when using Carafe with fine-tuned libraries versus libraries generated using pretrained DDA models. Regarding the impact of training data size on fine-tuning, this has been investigated in Figures 3–4 of the original AlphaPeptDeep study [3]. The peptides detected from a single DDA run from a human or yeast sample provide ample training examples to achieve robust fine-tuning.

3. I assume the pre-trained AlphaPeptDeep model supports different (N)CE settings. Please provide evidence that a much less resource-intensive calibration of the NCE does not yield similar results for e.g. the Exploris data - for each precursor charge and possibly offset to the center of the isolation window do account for differences in the way how NCE is calculated between DDA and DIA.

We have followed the reviewer’s suggestion to compare the libraries generated with MS2 fine-tuning and the libraries generated following the NCE calibration procedure used in a recent study [4]. These results suggest that fine-tuning of fragment ion intensity predictions is more effective than NCE calibration. We have added the new analysis to the revised manuscript in the result section “Carafe increases the number of peptides detected with DIA-NN” as shown below:

We also compared the performance of fine-tuning of fragment ion intensity prediction with normalized collision energy (NCE) calibration (see Methods). In brief, following the NCE calibration procedure in a recent study [4], for a given dataset, a spectral library was generated using the pretrained DDA model from AlphaPeptDeep with the optimal NCE determined using the corresponding training data. As shown in Supplementary Figure R4, 3.7–9.1% more precursors were

detected using fine-tuning than NCE calibration on three DIA datasets. These results suggest that fine-tuning of fragment ion intensity predictions is more effective than NCE calibration.

Figure R4: **NCE calibration performance evaluation.** (a), (c) and (e) X axis shows the NCE values considered during the optimization. Y axis shows the unweighted spectral entropy values. The best NCE was determined by highest median entropy value and was highlighted with purple color. The NCE value used for generating a dataset from a Thermo instrument was highlighted as light blue. The NCE value used for the TripleTOF 5600 dataset by Carafe fine-tuning was highlighted as light blue. (b), (d) and (f) show the number of precursors detected passed 1% FDR using DIA-NN with different spectral libraries. DDA: a spectral library generated using the pretrained DDA model from AlphaPeptDeep with the NCE used for generating the dataset or a default of 27 for the TripleTOF 5600 dataset. NCE calibration: a spectral library generated using the pretrained DDA model with the optimized NCE. MS2 fine-tuning: a spectral library generated using Carafe with only fragment ion intensity prediction fine-tuned. For boxplots, centerline indicates the median, the lower and upper hinges correspond to the first and third quartiles, whiskers indicate the 1.5 interquartile range and points indicate outliers.

4. In Figure 2, “TripleTOF 5600, SWATH” is listed 3 times, with varying improvements as a result of fine-tuning. However, the “n” is the same, which I find unlikely to happen if different datasets are plotted.

The same number n of peptides from the same dataset (TripleTOF 5600, SWATH) were used for Figure 2c–g. Note that we did not select the value of n ; rather, we used all the peptides returned by the training data processing. In these plots, we tested different models on the same test data. This is why the n is the same for all three plots. We have added a sentence to the figure legend to clarify this in the revised manuscript.

5. In Figure 2, the axis labels are unclear and do not specify what is being plotted. This information is only provided in the legend. The authors should add proper axis labels for clarity.

We have added axis labels for clarity in the revised manuscript, as shown below:

Figure R5: **Fine-tuning with DIA data improves fragment ion intensity prediction.** (a–d) Fragment ion intensity prediction performance comparison between a pretrained DDA model (x-axis) and models trained using DIA data on four different datasets (y-axis). **SPC: Spearman correlation.** (e–g) The performance of the shared peak detection methods in fragment ion intensity prediction. Four models were trained using the same human training data but with different peak masking strategies: no masking, spectrum-centric masking, peptide-centric masking, or both strategies. **The models were evaluated on the same test TripleTOF 5600 yeast DIA dataset. The test data were generated using the same process with the training data.** Each panel compares the Spearman correlation between observed and predicted fragment ion intensities for two different masking strategies. Each dot represents a peptide precursor (peptide sequence + modification + precursor charge state). **In each plot, n is the number of precursors used in evaluation.** The values are Spearman correlations between observed fragment ion intensities and predicted intensities. Only peaks determined to be non-interfered were used in the correlation calculation. The percentages on each plot indicate the proportion of spectra for which one method achieves a higher or lower correlation than the other. (h) The proportion of shared peaks detected on the TripleTOF 5600 DIA dataset. “*”: staggered isolation window.

6. The plotting of Figure 2 could be improved by using a 2D density. The current visualization does not highlight the majority of the data points.

As suggested, we have modified Figure 2 to use a 2D density plot.

7. For the peptide-centric and spectrum-centric masking approaches, the authors should provide information on the average number of peaks masked compared to the original number of peaks. This would help assess whether the majority of peaks are masked and how this affects the selection of correct identifications.

We have shown that the percentage of peaks determined as shared using the peptide-centric and spectrum-centric methods on the TripleTOF 5600 dataset. These results are provided in Figure 2h and Supplementary

Figure 5.

8. In Figure 4, the authors present the numbers of identifications but do not illustrate the overlap between different methods. It is important to demonstrate whether the identifications made with the new retrained models are shared with other methods or are entirely novel.

Following the reviewer's suggestion, we have added upset plots (Supplementary Figure S7–10) to show the overlap between different methods as shown below.

Figure R6: **Upset plot on the TripleTOF 5600 dataset.** The dots representing combination sets are colored according to the number of sets included.

Figure R7: **Upset plot on the metaproteomics DIA dataset.** The dots representing combination sets are colored according to the number of sets included.

Figure R8: **Upset plot on the metaproteomics phosphoproteomics DIA dataset.** The dots representing combination sets are colored according to the number of sets included.

Figure R9: **Upset plot on the metaproteomics DIA dataset Lumos re-CID dataset.** The dots representing combination sets are colored according to the number of sets included.

Remarks on code availability:

No documentation on source-code level. Github only contains the “first release”, no history of the developed code, implying that the code was/is developed somewhere else.

The project was started as a small piece of a larger effort and then later separated out. Hence, the early history of the code development is not available in the current Github repository.

Most (if not all) logic is in AIGear.java, which does not follow good coding practices.

We will improve the code in the next release.

Not a fan of the duplication of the AlphaPept code base, which requires the authors to keep their copy up to date as well now.

We do not intend to update Carafe with future versions of the AlphaPeptDeep code. Duplicating the code here removes an external dependency which might break in future versions.

Reviewer #2

In this study, Bo Wen et al. developed Carafe, a tool to predict high-quality spectral libraries for DIA analysis. Unlike conventional approaches relying on DDA data, this tool used DIA data to fine-tune a pre-trained model (provided by AlphaPeptDeep, in this case). This approach was shown to generate better spectral libraries, featured by more accurate fragment ion intensities and retention times. The concept of integrating information extracted directly from DIA data into spectral library generation is undoubtedly appealing. It has the potential to improve the re-analysis of DIA data itself or the data from similar MS settings. Furthermore, building an interference-free spectral library is essential to avoid misleading peptide quantification caused by shared fragment ions. Overall, I think the proteomics community would benefit from this tool and its underlying idea. To further improve the quality of the manuscript, I wish the authors may consider to address the following points:

We thank the reviewer for this accurate summary of our contributions.

1. Most results in the paper were about qualitative analysis – carafe was shown to outperform other methods in terms of the number of precursors identified. DIA-MS is particularly valued for its strength in quantitative proteomics. The manuscript could benefit from comparative analysis using quantitative metrics such as quantitative accuracy, relative errors, and coefficients of variation (CV) among replicates. Several DIA benchmarking datasets, such as those in PMID: 36609502 and PMID: 27701404 have been generated for such purpose by mixing proteins with known concentrations and could be utilized to benchmark the tool’s performance quantitatively. Demonstrating improvements in these metrics would provide stronger evidence of Carafe’s utility.

We have followed the reviewer’s suggestion to perform a quantitative evaluation using the LFQBench dataset (PMID: 27701404). We have added a new subsection of Results to describe this experiment, as shown below. We observed that the quantification accuracy and precision using the Carafe fine-tuned spectral library is comparable to both the DDA and the DIA-NN libraries.

2.5 Quantification evaluation

To evaluate the quantification precision and accuracy of using Carafe fine-tuned spectral libraries, we analyzed the LFQBench dataset [5] using Carafe’s fine-tuned library and four other spectral libraries. As shown in Figure R2a, using the Carafe fine-tuned library we identified 21.45% more precursors compared with the library generated using the pretrained DDA models and 10.02% more precursors compared with the library generated using DIA-NN’s built-in model. These results are consistent with our previous analysis, suggesting that both the fragment ion intensity and RT fine-tuning are useful for improving the number of precursors detected.

Next, we assessed quantification precision by comparing the Carafe fine-tuned library with two other libraries: one generated using the AlphaPeptDeep pretrained DDA model (hereafter “DDA”) and the other generated using the DIA-NN built-in model (hereafter “DIA-NN”). We calculated the coefficient of variation (CV) for precursor intensities across three replicates for each sample (samples A and B). As illustrated in Figure R2b, the Carafe fine-tuned library exhibited slightly lower median CVs, indicating improved precision at the precursor level relative to the DDA and DIA-NN libraries.

We next compared the quantification accuracy using Carafe’s fine-tuned library versus the other two spectral libraries. We compared the estimated precursor ratios between sample A and sample B to their expected ratios (1:1 for human, 10:1 for yeast and 1:10 for *E.coli*). As shown in Figures R2c–e, we observed that the quantification accuracy using the Carafe fine-tuned spectral

library is comparable to both the DDA and the DIA-NN libraries.

2. One significant contribution of Carafe is its ability to detect interference peaks arising from shared fragment ions. However, the manuscript does not clearly demonstrate the downstream benefits of removing such misleading signals. The authors could leverage gold-standard DIA datasets to illustrate these advantages. For example: How much do relative errors decrease when interference is removed? How do different strategies for determining shared peaks compare in terms of their impact on accuracy?

We agree with the reviewer that one significant contribution of Carafe is its ability to detect interference among peaks arising from shared fragment ions. When we fine-tune the fragment ion intensity prediction model in Carafe, we mask out those interfered peaks to improve the performance of fragment ion intensity prediction and then use the fine-tuned model to predict fragment ion intensity for spectral library generation. Note that peak masking is only applied during model training, not during prediction. We have shown that peak masking is useful for improving fragment ion intensity prediction by comparing (1) a fragment ion intensity prediction model training using peak masking, (2) a model trained without peak masking and (3) models trained with peak masking using one of the shared peak detection strategies, as shown in Figure 2e-f. To further investigate the impact of peak masking on peptide detection, during the revision we have performed a new analysis to compare a spectral library generated using a fragment ion intensity prediction model fine-tuned without peak masking with the library generated using peak masking. On the TripleTOF 5600 dataset, more peptide precursors were detected using the library with peak masking. We have described the new analysis in the section “Carafe increases the number of peptides detected with DIA-NN” as shown below:

To further investigate the utility of fine-tuning of fragment ion intensity predictions, we generated a spectral library using Carafe without peak masking on the TripleTOF 5600 dataset. A total of 35,134 precursors were detected using DIA-NN with this library, which is 2% lower than the number of precursors (35,837) detected with the library generated using Carafe with peak masking.

3. For the phosphoproteome analysis, the authors may consider validating Carafe using synthetic phosphoproteome datasets, particularly focusing on phosphosite localization accuracy. Ideally, an interference-free library or one built with discriminative fragment ions is expected to help with PTM site localization and quantification of peptides with PTMs. Including such analyses would broaden the tool’s applicability, especially for PTM data.

We have followed the reviewer’s suggestion to validate Carafe using a yeast phosphoproteome DIA dataset generated with a set of spiked in, synthetic phosphopeptides, as described below:

To further validate the quality of the spectral libraries generated by Carafe for phosphoproteome DIA data analysis, we estimated the false localization rates when analyzing a yeast phosphoproteome DIA dataset generated with a set of spiked in, synthetic phosphopeptides using DIA-NN with different spectral libraries (Figure R3g, Methods). As shown in Figure R3h, using stringent cutoffs ($PTM.Q.Value \leq 0.01$ and $PTM.Site.Confidence \geq 0.75$), we identified 27.7% more phosphopeptides with Carafe’s fully fine-tuned spectral library than with the library generated by DIA-NN. Consistent with our previous analysis, the fully fine-tuned Carafe library outperformed the libraries generated using the pretrained DDA model and the partially fine-tuned models. To evaluate the false localization rate, we calculated the proportion of detected synthetic phosphopeptide precursors with incorrect site localization. As shown in Figure R3i, the false localization

rate was 0.0% using the fully fine-tuned Carafe library, whereas the false localization rate was 1.3% using the library generated by DIA-NN. Similar results were observed using a different replicate from the same experiment (Supplementary Figure S15). These results provide evidence that Carafe is capable of generating high quality spectral libraries for phosphoproteome DIA data analysis.

4. ROC curves and comparisons at the protein level may be considered to be added.

In our recent study [2], we have evaluated FDR control evaluation of three widely used DIA tools, including DIA-NN, using an entrapment strategy on a wide range of DIA datasets. One of the major findings of that study is that the FDR control of these tools typically fails at the protein level. Therefore, in the present study, we have focused on the analysis at the precursor or peptide level. During the revision, we have added a ROC curve at the precursor level (Supplementary Figure S13). The ROC curve shows that the trend we observed in Figure 4 is consistent across different FDR thresholds.

Figure R10: The number of precursors detected using DIA-NN with different spectral libraries as a function of qvalue threshold on the TripleTOF 5600 dataset. DDA: spectral library generated using the pretrained DDA model from AlphaPeptDeep; MS2: spectral library generated using a fine-tuned fragment ion intensity prediction model but the pretrained RT model from AlphaPeptDeep; RT: spectral library generated using a fine-tuned RT prediction model but the pretrained fragment ion intensity model from AlphaPeptDeep; MS2/RT: spectral library generated using fine-tuned fragment ion intensity and RT prediction models; DIA-NN: spectral library generated using DIA-NN.

5. The authors may consider giving more explanation or illustration. For example, figure 1b and 1c seem not clear enough (e.g. peaks across DIA and DDA spectra were not aligned; the numbers in the tables seem to be random and are difficult to interpret...). The authors are encouraged to refine these figures to make them more intuitive and easier to follow. More real-world examples (as in the format of Figure 1c) could be added to explicitly showcase the advantages of removing shared peaks using complementary strategies.

We have improved the figure to make the DDA spectrum in Figure 1b align with the DIA spectrum in the m/z dimension. Due to limited space, it is difficult to use a real example for the illustration. Therefore, we have not changed the table in the plot. Instead, we have added a few sentences to the figure legend to explain the numbers in the table and the figure as shown below:

Figure R11: **Overview of Carafe.** (a) Using Carafe to generate an experiment-specific *in silico* spectral library for DIA data analysis. (b) The difference in fragment ion intensity model training between using DDA data (top) and using DIA data (bottom). The numbers in the tables represent illustrative (not real) observed (left) or predicted (right) fragment ion intensities, normalized to a range of 0 to 1. Each column represents a different type of fragment ion with a specific charge state (e.g., “b+” in the first column means a b-ion with charge 1+). Each row corresponds to the fragment ions generated by the fragmentation of a peptide at specific peptide bonds between amino acids (e.g., the first row in the first column is the b1 ion). (c) Shared peak determination in Carafe: the spectrum-centric approach (left) and the peptide-centric approach (right). The former determines shared fragment ions by identifying fragment ions in a single MS2 spectrum that are matched to multiple detected peptides. The latter determines shared peaks through by the correlation of the extracted ion chromatograms of fragment ions from the same peptide. The two methods are complementary and are performed independently. The shared peaks determined by the two methods are combined, and that information is used for peak masking during fragment ion intensity prediction model training in Carafe. CE: collision energy. NCE: normalized collision energy.

We have also followed the reviewer’s suggestion to add a few real-world examples to showcase the advantages

of removing shared peaks using complementary strategies. We have added the new figures to the revised manuscript as Supplementary Figures as shown below:

Figure R12: **A shared peak detection example using the two-pronged approach in Carafe.** (a) The top panel shows the observed fragment ions for peptide GVLNALAGR with precursor charge 2+. The fragments highlighted as red color are shared fragments determined using Carafe. The top panel shows the predicted fragment ion intensity values using the pretrained AlphaPeptDeep model. (b) Extracted ion chromatograms (XICs) of all observed fragment ions matched to peptide GVLNALAGR. The fragments highlighted as red color are shared fragments determined using Carafe. The dashed lines indicate the peak boundaries. (c) A different peptide YNIEKDIAAHIK with precursor charge 3+ was detected from the same MS2 spectrum that was used to detect the peptide GVLNALAGR showing in (a). Only observed fragment ions matched to the two peptides were shown. The fragment ions annotated to both peptides are highlighted as red color. (d) XICs of the observed fragment ions matched to peptide GVLNALAGR which were determined as unshared peaks using the peptide-centric method by Carafe. The fragment highlighted as red color (b9) was only determined as a shared peak by the spectrum-centric method by Carafe. This fragment was not determined as a shared peak using the peptide-centric method due to its high correlation (Pearson correlation: 0.91) to the best fragment ion XIC of the peptide.

Figure R13: **Examples of shared peak detection by Carafe on the TripleTOF5600 dataset.** (a), (c) and (e) show the observed fragment ions and predicted fragment ion intensity values for the three peptides. The fragments highlighted as red color are shared fragments determined using Carafe. (b), (d) and (f) show the XICs of all observed fragment ions matched to the corresponding peptides. The fragments highlighted as red color are shared fragments determined using Carafe. The dashed lines indicate the peak boundaries.

Figure R14: Examples of shared peak detection by Carafe on the Exploris 480 metaproteome dataset. (a), (c) and (e) show the observed fragment ions and predicted fragment ion intensity values for the three peptides. The fragments highlighted as red color are shared fragments determined using Carafe. (b), (d) and (f) show the XICs of all observed fragment ions matched to the corresponding peptides. The fragments highlighted as red color are shared fragments determined using Carafe. The dashed lines indicate the peak boundaries.

Figure R15: **Examples of shared peak detection by Carafe on the Astral dataset.** (a), (c) and (e) show the observed fragment ions and predicted fragment ion intensity values for the three peptides. The fragments highlighted as red color are shared fragments determined using Carafe. (b), (d) and (f) show the XICs of all observed fragment ions matched to the corresponding peptides. The fragments highlighted as red color are shared fragments determined using Carafe. The dashed lines indicate the peak boundaries.

6. Figures 2 and 3 highlight Carafe’s improvements. However, the manuscript would benefit from including additional concrete results, such as a few gained peptides with corresponding predicted spectra and real spectra with annotated peaks.

We have followed the reviewer’s suggestion to add a few gained peptides with corresponding predicted spectra and real spectra with annotated peaks. In the revised manuscript, we have added the new figures and a sentence to mention this to the section “Carafe increases the number of peptides detected with DIA-NN,” as shown below.

A few of the peptides gained by using Carafe’s fine-tuned spectral libraries with corresponding predicted spectra and observed spectra with annotated peaks are shown in Supplementary Figures R16–R17.

Figure R16: Annotated spectra of four peptides detected using Carafe fine-tuned spectral libraries but not detected using the pretrained model (AlphaPeptDeep) derived spectral libraries. For each subplot, the top panel shows the experimental spectrum while the bottom panel shows the predicted spectrum. Only observed fragment ions matched to the corresponding peptide were shown.

Figure R17: Annotated spectra of two phosphopeptides detected using Carafe fine-tuned spectral libraries but not detected using the DIA-NN generated spectral library. For each subplot, the top panel shows the experimental spectrum while the bottom panel shows the predicted spectrum. Only observed fragment ions matched to the corresponding peptide were shown.

7. While the manuscript is understandable, the paper, if possible, can be edited again by a native English

speaker.

We have revised the manuscript to improve the grammar and usage.

References

- [1] Noah M Lancaster, Pavel Sinitcyn, Patrick Forny, Trenton M Peters-Clarke, Caroline Fecher, Andrew J Smith, Evgenia Shishkova, Tabiwang N Arrey, Anna Pashkova, Margaret Lea Robinson, et al. Fast and deep phosphoproteome analysis with the orbitrap astral mass spectrometer. *Nature Communications*, 15(1):7016, 2024.
- [2] Bo Wen, Jack Freestone, Michael Riffle, Michael J MacCoss, William S. Noble, and Uri Keich. Assessment of false discovery rate control in tandem mass spectrometry analysis using entrapment. *Nature Methods*, 2025. In press.
- [3] Wen-Feng Zeng, Xie-Xuan Zhou, Sander Willems, Constantin Ammar, Maria Wahle, Isabell Bludau, Eugenia Voytik, Maximillian T. Strauss, and Matthias Mann. Alphapeptdeep: a modular deep learning framework to predict peptide properties for proteomics. *Nature Communications*, 13(1):7238, 11 2022.
- [4] Ludwig Lautenbacher, Kevin L Yang, Tobias Kockmann, Christian Panse, Matthew Chambers, Elias Kahl, Fengchao Yu, Wassim Gabriel, Dulguun Bold, Tobias Schmidt, et al. Koina: Democratizing machine learning for proteomics research. *bioRxiv*, 2024.
- [5] P. Navarro, J. Kuharev, L. C. Gillet, O. M. Bernhardt, B. MacLean, H. L. Rost, S. A. Tate, C. C. Tsou, L. Reither, U. Distler, G. Rosenberger, Y. Perez-Riverol, A. Nesvizhskii, R. Aebersold, and S. Tenzer. A multicenter study benchmarks software tools for label-free proteome quantification. *Nature Biotechnology*, 34(11):1130–1136, 2016.

Reviewer #1

I commend the authors for the work they put into the response and am happy to see most points addressed. I am particularly impressed by the integration in Skyline which boost applicability and value of the proposed approach and the new data and analysis added. However, two major and two minor points remain to be addressed.

Thank you for the positive comments on our previous revision of the manuscript.

Major Points:

1. On the importance of masking, the authors show that a 2% increase can be achieved when fine-tuning a model with masking compared to fine-tuning without masking. This is a rather marginal improvement and appears to support my point that masking is likely not as important as the authors make it appear in the paper (2 Figures + a large body of text). I would like to see a more systematic evaluation of this in the manuscript and depending on the results an adjustment of wording (wrt novelty and importance of this) by doing the analysis the authors did at least 3 times (technical replicates) for masking and no masking for more than just one example (as best covering different MS instruments that are “close” or “far away” from the original model). The example the authors choose could be an outlier since TOF data tends to be more noisy compared to Orbitraps and masking may e.g. largely remove noise from the training. At this point, it is not clear whether the 2% gains are likely to be observed for other MS instruments and to which degree the 2% gains are a result of (partially) non-deterministic algorithms (both the fine-tuning and I assume DIA-NN are not deterministic). Judging by eye, Figures R6 and R7 show that there is quite a number of unique precursors confidently identified by each of the 5 evaluated methods, that may as well represent the 2%. I understand the reasoning that masking may potentially be important when attempting to fine-tune models based on more complex DIA data, however, as far as I can tell right now, this is not supported with evidence, yet. Such a claim may be substantiated if the increase observed when comparing masked vs non-masked fine-tuning correlates with the “complexity” of the DIA data it was trained on, as stated by the authors by the discussion.

As requested, we have now repeated the analysis on the other three datasets used in Figure 4. These datasets were generated from two different MS instruments which are different from the TripleTOF 5600 dataset. Comparing the fine-tuned model with masking to the fine-tuned model without masking, the improvement varies across different datasets, but fine-tuning with masking always increased the sensitivity of peptide detection. Also, we agree with the reviewer that fine-tuning is probably most beneficial for noisy data and that masking will add little value in the absence of interference. The improvement in peptide detection may depend on both the degree of interference of the training and the complexity of the data. Given that TOFs, including the Sciex TripleTOF/ZenoTOF, Bruker TimsTOF, or Thermo Astral (a multireflectron TOF), are becoming the standard for DIA experiments, we believe that fine-tuning with masking has value because Orbitraps will be used less and less for DIA experiments in the future. Please note that for all four datasets in Figure 4, the datasets used for model fine-tuning were all generated from human cell line samples, so the sample complexity is similar. We have added the new analysis to the revised manuscript in the section “Carafe increases the number of peptides detected with DIA-NN” as shown below:

To further investigate the utility of fine-tuning of fragment ion intensity predictions, we generated a spectral library using Carafe without peak masking on the TripleTOF 5600 dataset. A total of 35,134 precursors were detected using DIA-NN with this library, which is 2.0% lower than the number of precursors (35,837) detected with the library generated using Carafe with peak masking. We repeated this analysis on the three other datasets used in Figure 4 and found that the number of precursors detected using DIA-NN with the library generated without peak masking was 0.6% lower on the phosphoDIA dataset, 2.4% lower on the reCID dataset, and 7.5%

lower on the metaproteome dataset. These results suggest that peak masking provides a modest but consistent improvement in peptide detection.

In addition, we also updated the main text accordingly to de-emphasize the importance of the masking strategy, including the description of the masking strategy in Results:

Peaks with interference are often highly abundant because multiple fragment ions contribute to their intensities (Supplementary Figures S1–S4). The degree of interference in DIA spectra is often unknown and varies across experiments. Accordingly, the effect of training with interfered peaks is difficult to predict a priori. Therefore, masking out such peaks during training could help mitigate their potential adverse impact on performance.

and the corresponding discussion in the Discussion:

Fine-tuning on DIA data for *in silico* library generation raises the issue of managing shared fragment ions when constructing training data for fragment ion intensity prediction. To address this, we apply an interference peak detection method used for training data generation and a peak masking strategy used in model training to handle naturally chimeric spectra in DIA data. Although the benefit of masking may be modest under conditions where the true fraction of shared peaks tends to be low and the complexity of the MS data under analysis is not high, the actual degree of interference in DIA spectra is typically unknown and can vary substantially from one experiment to another. Consequently, training on unmasked, interfered peaks carries unpredictable effects on model performance, making it advisable to include explicit interference detection and peak masking during model training to ensure robust predictions across diverse DIA LC-MS/MS setups.

2. I am aware that “peptide detection in DIA workflows typically involves multiple complex steps, and the search tool that we used in this analysis is not open source”. However, I politely disagree with the consequence that one should not attempt to understand why an increase in SPC of only 0.05 leads to substantial gains in identified precursors. To me, and I would assume most readers as well, the reported improvement in prediction accuracy appears very small in relation to the reported gains. I suggest that the authors attempt to investigate the relation between increased SPC and observed gains. For example, the authors could check whether the precursors gained are cases of formerly low but now high SPC (after fine-tuning), whether those show a substantially bigger improvement compared to other precursors (e.g. the shared or lost). Are the lost peptides largely the ones which have lower SPC after fine-tuning? Do the gained peptides show a similar increase in SPC as the peptides in the test-set? Related to this is whether the authors would expect to see another increase in IDs if a different fine tuning method would result in an additional boost in performance of on average 0.05 SPC?

We have followed the reviewer’s suggestion and investigated the relationship between SPC and the number of precursors identified. We found that the precursors gained by fine-tuning were indeed those with low SPC in the pre-trained model, and that these precursors showed a larger increase in SPC after fine-tuning than the shared or lost precursors. The SPC improvement for the shared precursors was similar to that on the test dataset. We have added this analysis to the revised manuscript in the section “Carafe increases the number of peptides detected with DIA-NN” as shown below, including a new supplementary figure (Figure S15) that illustrates these findings.

To investigate the utility of fine-tuning of fragment ion intensity predictions, we compared the precursors detected using the Carafe fine-tuned library (referred as “MS2”) and the precursors detected using the spectral library generated without fine-tuning (referred as “DDA”) on the TripleTOF 5600 dataset. As shown in the Supplementary Figure R1, we found that the precursors gained by fine-tuning were those with low Spearman correlation in the pre-trained model, and that these precursors showed a larger increase in Spearman correlation after fine-tuning than the shared or lost precursors. The median Spearman correlation improvement (0.05) for the shared precursors was similar to that on the test dataset we showed previously.

Figure R1: **Spearman correlation improvement analysis for precursors detected on the TripleTOF 5600 yeast dataset.** (a) Spearman correlation comparison for the precursors detected by both the Carafe fine-tuned spectral library (only fine-tuned fragment ion intensity prediction model, referred as “MS2” in Figure 4) and the spectral library generated from the AlphaPeptDeep pre-trained model (referred as “DDA” in Figure 4). (b) Same as (a) but for the precursors only detected using the “MS2” library. (c) Same as (a) but for the precursors only detected using the “DDA” library. SPC: Spearman correlation. In each plot, n is the number of precursors used in analysis. The value in the second row is median Spearman correlation improvement comparing the predictions using Carafe fine-tuned model with those using the AlphaPeptDeep pre-trained model. Only peaks determined to be non-interfered were used in the correlation calculation.

Minor Points:

1. While the authors show some example spectra of improved performance of the fine-tuned model (Figures S11 and S12), the examples picked appear to be (without exception) extreme examples for improved performance. The average, as depicted in Figure S6, is much lower than the (estimated) average 0.4 improvements shown. In my view, this leaves an incorrect impression about the performance of fine-tuning and the authors should show examples of at least average performance as well.

We have followed the reviewer’s suggestion and added a new figure (Supplementary Figure R2) and added two more examples to the original Figure S12 (Supplementary Figure R3) that show examples closer to the

average performance of the fine-tuned model.

Figure R2: Annotated spectra of four peptides detected using Carafe fine-tuned spectral libraries but not detected using the pretrained model (AlphaPeptDeep) derived spectral libraries. For each subplot, the top panel shows the experimental spectrum while the bottom panel shows the predicted spectrum. Only observed fragment ions matched to the corresponding peptide are shown.

Figure R3: Annotated spectra of four phosphopeptides detected using the Carafe fine-tuned spectral library but not detected using the DIA-NN generated spectral library. For each subplot, the top panel shows the experimental spectrum while the bottom panel shows the predicted spectrum. Only observed fragment ions matched to the corresponding peptide are shown.

2. The CE calibration plots added (Figure S14) show an unexpected characteristic. One would have expected a more substantial difference between e.g. NCE 20, 29, and 40. However, the spectral entropy shows almost no difference along NCE. I encourage the authors to double check if the e.g. input to the model was provided correctly. In other examples one can find online, the difference in prediction performance is reported to be much higher when choosing the incorrect NCE. This may suggest that the pre-trained model is not as aware of the impact of NCE as reported by other tools and thus the results reported here are not representative of what NCE calibration may achieve. This may warrant a brief mention in the discussion.

We have double-checked the inputs to the model and confirmed that it was provided correctly. We agree that the spectral entropy does not show a substantial difference as a function of NCE among the three datasets we tested. We have added the following sentences to the Discussion:

We also show that fine-tuning of fragment ion intensity predictions is more effective than a less

resource-intensive NCE calibration across different datasets. Notably, we do not observe substantial differences in spectral entropy across different NCE values. This trend likely reflects the fact that while the pretrained model was trained on data from multiple instruments, only one type of instrument (Lumos) contributed data with varying NCEs during training. However, the datasets used in our NCE analysis were acquired on different instrument types that were not represented in the NCE-diverse training data. Consequently, the model may be less sensitive to NCE variations for these specific instrument configurations, potentially limiting the representativeness of our NCE calibration results.

Remarks on code availability:

Despite the authors stating to improve the code for the next release, which is now in release 1.0.0, no changes were made. Most (if not all) logic is in `AIGear.java`. No source code documentation provided.

Sorry that we didn't make more clear what changes we had made to the code. Before we released version 1.0.0, we mainly focused on improving the logic in the main class in `AIGear.java` so that it could be easily extended to support fine-tuning with both DDA data and TIMS-TOF DIA data, as well as supporting fine-tuning with different types of input data, such as inputs from Skyline or different versions of DIA-NN. In addition, we improved the logic associated with handling modifications. For this revision, we have added more than 740 lines of comments to the code, including annotations to all class variables and functions in the code in `AIGear.java`. Regarding restructuring the code logic, we are reluctant to embark on major refactoring at this stage due to the risk of introducing bugs into the code, which is now fairly well tested.

Reviewer #2

Overall, I think the authors did excellent work addressing all of my comments, and the manuscript can now be accepted for publication.

Thank you very much for this positive assessment of our work.

Some suggestions regarding writing and formatting:

1. In Figure S1 of the supplementary materials, "The top panel shows..." should be revised to "The bottom panel shows the predicted fragment ion intensity values using the pretrained AlphaPeptDeep model."

We have fixed this typo in the revised manuscript.

2. In Figure 5, it sees that "Panel g was generated using BioRender"

We have fixed this in the revised manuscript.

3. In Figure 7, the phrase "Precursor level" appears consistently across panels a-e. As no comparison was performed at the peptide/protein level, this phrase is redundant and can be removed.

Indeed, some clarification on comment #4: My previous suggestion about "ROC curves and comparisons at the protein level" specifically referred to counting the numbers of proteins, not controlling the FDR at the protein level.

As suggested, we have removed the phrase "Precursor level" from panels a-e in Figure 7.

4. References 26 and 41 appear to be duplicates; please check.

We have removed the duplicate references in the revised manuscript.